# The vertical position of visual information conditions spatial memory performance in healthy aging

Marion Durteste [1], Louise Van Poucke[1], Sonia Combariza[1], Bilel Benziane[1], José-Alain Sahel [1,2,3,4], Stephen Ramanoël [1,5,6 ✉] & Angelo Arleo [1,6 ✉]

Memory for objects and their location is a cornerstone of adequate cognitive functioning across the lifespan. Considering that human visual perception depends on the position of stimuli within the visual field, we posit that the position of objects in the environment may be a determinant aspect of mnemonic performance. In this study, a population of 25 young and 20 older adults completed a source-monitoring task with objects presented in the upper or lower visual field. Using standard *Pr* and multinomial processing tree analyses, we revealed that although familiarity-based item memory remained intact in older age, spatial memory was impaired for objects presented in the upper visual field. Spatial memory in aging is conditioned by the vertical position of information. These findings raise questions about the view that age-related spatial mnemonic deficits are attributable to associative dysfunctions and suggest that they could also originate from the altered encoding of object attributes.

[1] Sorbonne Université, INSERM, CNRS, Institut de la Vision, 17 rue Moreau, F-75012 Paris, France. [2] CHNO des Quinze-Vingts, INSERM-DGOS CIC, 1423 Paris, France. [3] Fondation Ophtalmologique Rothschild, Paris, France. [4] Department of Ophthalmology, The University of Pittsburgh School of Medicine, Pittsburgh, PA, USA. [5] Université Côte d'Azur, LAMHESS, Nice, France. [6] These authors contributed equally: Stephen Ramanoël, Angelo Arleo. ✉email: stephen.ramanoel@univ-cotedazur.fr; angelo.arleo@inserm.fr

To guide their behavior in everyday life, humans must understand, attend to, and interact with the visual information in the surrounding environment. In that respect, memory for objects (i.e., item memory) and their location (i.e., spatial memory) is key to preserving cognitive performance across the lifespan. Indeed, impaired item and spatial mnemonic capacities negatively affect autonomy and quality of life in healthy aging. Previous work has reported these deficits to be rooted in alterations of high-order associative functions such as contextual binding and executive function[1–3]. We propose the view that mnemonic deficits in older age could also emerge from the deficient encoding of specific object features. Indeed, item and spatial memory are complex skills that require visual processing of objects' various colors, textures, sizes, and positions in space. Modified sensory function in older age may thus also contribute to the decline in memory for objects and their spatial context[4,5].

We set forth the idea that location in space may be a particularly relevant object attribute considering that human visual perception depends strongly on the position of stimuli in the visual field[6]. Fine visual performance decreases sharply when moving away from the center of vision towards the periphery. It can also be modulated by isoeccentric locations around the visual field. One example of this effect is the horizontal–vertical anisotropy: at fixed eccentricities, visual performance is better along the horizontal than the vertical meridian. A second notable example is the vertical meridian asymmetry (VMA), which reveals that visual ability differs between the upper and lower visual fields[7,8]. These asymmetries exist across a vast array of stimuli orientations, sizes, and luminance levels[9], and they appear resilient to endogenous, exogenous, and presaccadic attention[10–12]. Visual acuity and contrast sensitivity[9,13], temporal resolution[14], spatial attention[15], hue discrimination[16], and motion processing[17–20] are better performed in the lower visual field. A bias for the upper visual field is evident in experimental paradigms in which visual search[21,22], change detection[23], target identification[24–26], or categorical judgements[27] are implicated. Although it is clear that the inhomogeneities around the visual field are pervasive across a variety of perceptual tasks, whether they extend to cognitive function and, more specifically, to item and spatial memory remains poorly understood. A few studies showed that the HVA and VMA could influence visual short-term memory[28,29]. However, these studies used non-naturalistic stimuli in the form of gratings or blocks, and they only tested very short-term memory. More evidence is needed to test the possibility that spatial location could condition the precision with which the memory trace is encoded or retrieved.

Importantly, these vertical performance asymmetries evolve across the lifespan. A burgeoning field of research is examining how upper–lower visual field asymmetries emerge during childhood. A recent study compared the VMA for contrast sensitivity between children and young adults[30]. The authors showed that the VMA is a malleable property of the visual system that settles in beyond childhood. Another recent study concluded that perceptual asymmetries in infants reflect adaptations to typical spatial locations in the surrounding environment[31]. Regarding aging, data on upper–lower asymmetries are scarce and remain largely inconclusive. Although some research has reported that the VMA persists in late adulthood[32,33], other psychophysical and visual search studies have revealed that vertical performance asymmetries are modified in older age[34–36].

Here, we sought to determine whether changes in visual encoding in one vertical hemifield could modulate the mnemonic performance of young and older adults. The aim of the present study was threefold. First, it strove to characterize the influence of the vertical position of information on item and spatial memory; second, it attempted to compare such visual preferences between young and healthy older adult populations; third, it verified whether the area of participants' upper and lower visual fields could explain performance differences. To address these questions, we used a source-monitoring task that assessed item and spatial memory separately, and we performed kinetic perimetry to quantify the extent of individual visual fields. We computed the probabilities of remembering objects and their spatial location as a function of upper or lower visual field presentation positions using standard *Pr* analysis and multinomial processing tree (MPT) modeling. The two types of analyses are complementary as the first is commonly used in the literature and easily interpretable and the second, although more complex, has the advantage of making explicit the assumptions about the relationship between item and spatial memory. Based on the few existing studies, we hypothesized that the vertical position of information would play a role in older adults' mnemonic abilities.

## Methods

**Participants.** We recruited a sample of 52 participants (26 young and 26 older adults) from the SilverSight cohort[37] to take part in the experiment. The cohort study consists of healthy adults without any history of psychiatric or neurological disorders who have undergone an exhaustive battery of clinical (ophthalmological, auditory, vestibular) and neuropsychological tests. All participants showed normal performance on the Mini-Mental State Examination[38], the perspective-taking test[39], the Corsi block-tapping task[40], and the 3D mental rotation test[41] (Supplementary Table 1). Moreover, they had normal or corrected-to-normal eyesight. All participants were French speakers, and self-reported sex was obtained exclusively in French. The self-report asked for their "sexe", which refers to both biological sex and societal gender roles. Information on "genre", which in French refers specifically to societal gender roles, was not obtained. Of the total 52 participants, we excluded one young adult due to a neurological disease diagnosis subsequent to their participation and six older adults based on poor eye-tracking performance (see the Eye-tracking section for more details). Consequently, the analyses included a group of 25 young adults (mean age: 29.1 ± 4.2 years old; 14 female participants and 11 male participants) and 20 older adults (mean age: 75.5 ± 3.7 years old; 12 female participants and 8 male participants). Participants received a 30€ gift voucher as compensation. The Ethical Committee "CPP Ile de France V" (ID_RCB 2015-A01094-45, CPP N: 16122) approved the experimental procedures, and all participants provided written informed consent.

**Apparatus and setting.** We used PsychoPy v2020.2.10[42] on a Dell monitor with a 1280 × 1024-pixel resolution. The screen subtended 29° of visual angle in height and 36° of visual angle in width. Participants sat 57 cm from the monitor with their heads positioned on the chinrest of a head-mounted monocular eye-tracker (EyeLink 1000 Tower Mount, SR Research Ltd., Canada). We collected responses with a mini numeric keypad (KKmoon) positioned flat on a table in front of the participants. Young and older adults with visual corrections removed their glasses to perform the task because multifocal glasses could have biased performance in the upper or lower visual fields. Finally, the experiment took place in a quiet, dim-lit room at the participants' preferred time of day.

**Stimuli.** Stimuli consisted of 327 photographs of objects selected from the datasets of the mnemonic similarity task[43] (available at https://github.com/celstark/MST) and from the Massive Memory unique object image set[44] (available at http://konklab.fas.harvard.edu/#). For this experiment, we removed images of animals or

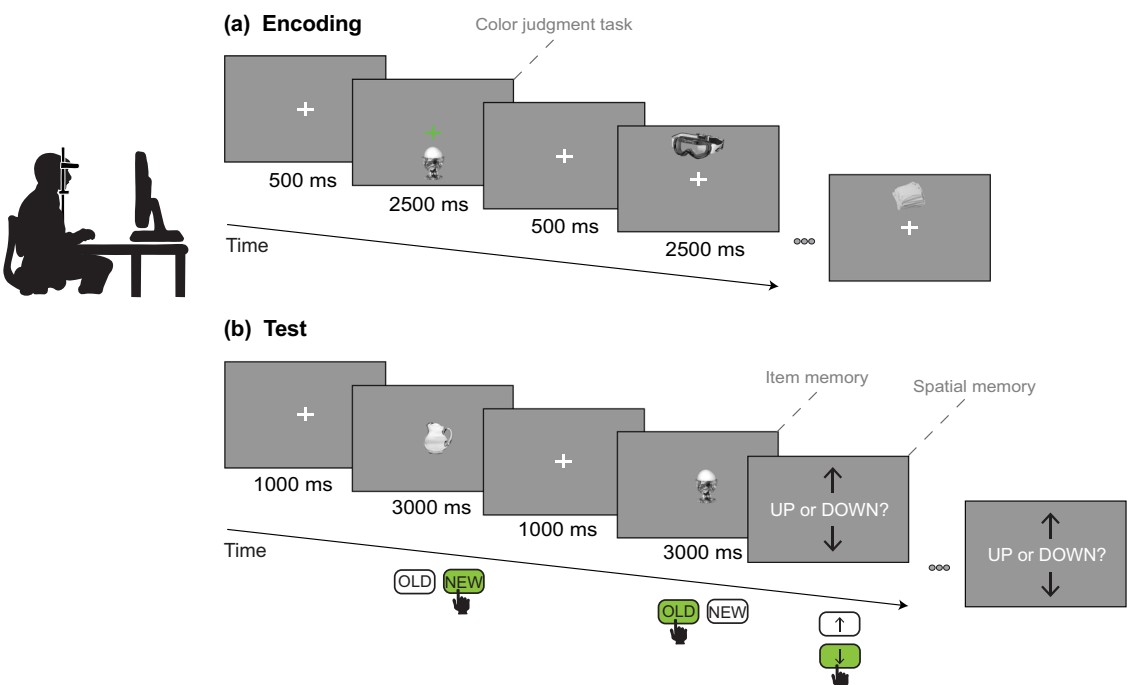

**Fig. 1 Schematic representation of the experimental paradigm.** Schematic depicting participants performing a desktop-based source-monitoring task while their gaze is recorded with an eye-tracker. Each block was divided into an encoding and a test phase. **a** During the encoding phase, participants saw 30 objects presented sequentially in the upper or lower part of the screen along the vertical meridian. They had to maintain fixation on the central cross throughout the duration of the phase. To facilitate fixation, they performed a color judgment task that consisted in pressing a key whenever the fixation cross turned green. **b** During the test phase, participants saw 20 objects that were previously presented and 10 new distractor objects. Participants first determined whether the object was old or new (item memory); if they thought that the item was old, they then decided whether the object appeared up or down (spatial memory).

people; of objects strongly associated with the ground, the ceiling, or the sky (e.g., plane, chandelier); and of uncommon objects (e.g., chisel, perimeter). We resized the selected images to a 400 × 400-pixel resolution and converted them to greyscale. Then, we normalized the images to obtain a mean luminance equal to 128 on a gray-level scale ranging from 0 to 255. The objects subtended ~4° of visual angle in height. Figure 1 presents example images of objects.

**Procedure and task design**. The experimental session began with a nine-point eye-tracking calibration. The eye-tracker sampled the position of the dominant eye at 1000 Hz throughout the entire session. We repeated the calibration halfway through the experiment. A 1.5-min practice run and eight experimental blocks each lasting ~4 min, structured the experimental session. Participants repeated the practice run until they understood the instructions and felt comfortable with the keypad. After completing four blocks, participants had a 5-min break during which they could remove their head from the chinrest. The session ended with a questionnaire about participants' strategies and subjective judgement of task difficulty.

We adapted the design of the paradigm from the classic source-monitoring paradigm that assesses both item recognition and source memory[45]. Each block comprised an encoding phase and a test phase (Fig. 1). The encoding phase started once participants had a stable fixation on the central cross for more than 800 ms. The task presented 30 unique objects sequentially, for 2500 ms each, in the upper or lower part of the screen along the vertical meridian. A 500 ms interval separated each stimulus presentation. Participants had to maintain fixation on the central cross throughout the entire duration of the encoding phase. To facilitate fixation, participants performed the additional task of

pressing a button when the cross turned green. During the test phase, participants saw 20 objects that had been presented during the encoding phase and 10 new objects (i.e., distractors). Only two-thirds of the encoded objects were presented during the test phase in order to limit task duration for older adults. Items were ordered randomly, but that order remained fixed across participants. Stimuli stayed in the center of the screen for 3000 ms, and participants decided whether the object was old or new by pressing the appropriate key (item memory). If participants responded that the item was old, we subsequently tested their memory of the object's position (spatial memory). Participants determined whether the object had appeared up or down in a self-paced manner. They didn't receive any feedback. A 1000-ms interval separated each trial in the test phase to leave enough time for older adults to reposition their fingers on the keypad. All timings were decided based on pilot testing with older participants and on previous studies using the source-monitoring paradigm.

To minimize order effects, we counterbalanced the sequence of blocks using a Latin square design. We also counterbalanced the studied objects by size and category across screen positions and across blocks. Finally, we counterbalanced the position of objects across participants. One-half of the participants saw *object 1* in the upper part of the screen, and the other half saw *object 1* in the lower part of the screen.

**Visual testing**. Trained orthoptists performed all visual testing. We assessed all participants' visual acuity and contrast sensitivity using the LogMAR scale and the Pelli-Robson score, respectively. We also performed kinetic perimetry on the Octopus 900 (Haag-Streit, Switzerland). We obtained participants' visual fields from both eyes using stimulus sizes of V1e, III1e, II1e, and I1e at 4

degrees per second for isopter charting. We adjusted participants' responses to stimuli presentation for reaction times. We then used custom MATLAB code to process the visual field data for each participant's left and right eyes. The latter extracted upper and lower visual areas along the horizontal meridian for the isopters corresponding to stimulus sizes of V1e, III1e, and II1e. We subsequently computed the vertical meridian asymmetry index as the difference in area between the lower and upper visual fields, divided by the mean of the two areas, and multiplied by 100:

$$VMA = \frac{(\text{lower visual field area} - \text{upper visual field area})}{\text{mean}(\text{lower visual field area, upper visual field area})} \times 100$$

Indices close to 0 indicate no asymmetry between the areas of the two hemifields. Lower indices reflect a vertical asymmetry with a larger area dedicated to the upper than to the lower visual field. In contrast, higher indices reveal a vertical asymmetry with a larger area for the lower visual field than for the upper visual field. We excluded two older adults from the visual field analyses because their kinetic perimetry data were extracted in a wrong format. The comparison between their data and those from other participants was thus impossible.

**Data analysis**. We processed the behavioral and eye-tracking data using custom Python code (Python version 3.8.1 in Spyder IDE version 4.1.5) and we conducted all statistical analyses using R version 4.0.3 in RStudio version 1.4.1103 (R Core Team, 2020; RStudio Team, 2021). We disclose that there was no pre-registration of the analysis plan for this research.

To classify gaze data into fixations, we chose the velocity-based algorithm implemented in the *saccades* R library[46]. Before analysis, we removed all fixations that lasted under 100 ms. We then used the 2/4 method and corresponding Fujii classification to assess fixation stability and identify participants who failed to maintain central fixation throughout the encoding phases[47,48]. The latter method considers participants to have unstable fixation if less than 75% of their fixation points are inside a 4° circle. As previously mentioned, we consequently excluded six older adults.

To assess participants' item and source memory, the literature uses a discrimination index or *Pr*, which is a measure of accuracy developed by Snodgrass & Corwin (1988)[49] for such paradigms. For item memory, the *Pr* was calculated by subtracting the proportion of false alarms from the proportion of hits. For spatial memory, the *Pr* was calculated by subtracting the number of incorrect spatial judgments from the number of correct spatial judgments and dividing by 160 (= the total number of source trials that there should be if a participant scored perfectly on the item memory task) in order to account for the interindividual variability in the number of source trials. If a participant frequently answered that an item was new, they would have far less source trials than a participant who frequently answered that an item was old. Reaction times were recorded for both item and spatial memory. They corresponded to the elapsed time between the presentation of the object and the button press. Two linear mixed models were used to evaluate the influence of age group and initial object position on item *Pr* and source *Pr*. The random-effects structure was chosen to best fit the experimental design while remaining parsimonious[50]. The two models thus contained participants as random intercepts. The reliability of each fixed effect was tested using the Akaike information criterion (AIC) goodness-of-fit statistic, comparing one model which included the effect of interest and another which didn't. Using this procedure, age group and object position were kept as fixed effects after determining that the inclusion of sex and block sequence did not improve model fits. The normality of residuals

of the two linear mixed models were carefully inspected using residual plots. Reaction time is positively skewed by nature. Following recent recommendations on how best to model such data[51], we conducted two gamma generalized linear mixed models with a log link function to analyze reaction times for the item memory and spatial memory tasks. In both models, the random-effects structure included participant and block as random intercepts along with the random effect of trial in each age group. The fixed effects were age group, object position, and block number. Sex was removed because it did not improve model fit.

We performed MPT modeling to distinguish item and spatial memory between objects presented in the upper part of the screen and objects presented in the lower part of the screen. By analyzing response frequencies, MPT models provide probability estimates of latent cognitive processes implicated in a specific paradigm[52,53]. Indeed, MPT models rely on extensive theoretical work, making them ideally suited to disentangle the meaningful psychological phenomenon implicated in the task at hand. In the source-monitoring paradigm, spatial memory indices are often biased by item recognition and guessing prevalence. Therefore, MPT measurements are particularly useful because they precisely isolate spatial memory[54,55]. The item-source model in which spatial memory is considered to be dependent upon item recognition has been the model of choice in most studies using MPT modeling for source-monitoring tasks. However, researchers recently put into question the theoretical assumptions behind this unidimensional account of memory[56]. The dual-process source-item model presents itself as an interesting alternative by suggesting that both spatial and item memory can be supported by a recollection mechanism, whereas item memory can also be subtended by familiarity alone when recollection fails[45,57]. Taking the latter into account, we chose to apply the source-item model to the present paradigm. According to the source-item model, $I_{up}$ and $I_{down}$ depict the phenomenon of familiarity that can occur when recollection fails for objects presented in the upper or lower visual fields. In contrast, $S_{up}$ and $S_{down}$ reflect the process of recollection in which both the object and its position are remembered simultaneously for objects that were presented in the upper or lower visual fields. The parameter o corresponds to the probability of guessing that the object was presented (i.e., answering that the item was old), and the parameter g estimates the probability of guessing that the item was presented in the upper part of the screen, given that item recognition has failed.

We fit the behavioral data to the MPT model for each age group separately with the *TreeBUGS* package in R[58]. This R package allows hierarchical models to be fit using Just Another Gibbs Sampler (JAGS) to estimate posterior distributions. Specifically, we used latent-trait hierarchical models because they take into account both interindividual variability and parameter correlation within participants[59]. We kept the weakly informative default priors provided by the *TreeBugs* package and included sex as a covariate in the MPT models. The models estimate the latent-trait parameters using Markov chain Monte-Carlo simulations in JAGS. The algorithm performed 50,000 iterations with a burn-in period of 10,000 samples and thinning factor of 10. $R < 1.05$ indicated that a parameter had reached good convergence. We relied on posterior predictive p-values $p_{T1}$ and $p_{T2}$ to evaluate the goodness-of-fit of the final models[59]. The $p_{T1}$ and $p_{T2}$ variables measure model fits in terms of means and covariance, respectively. Predictive p-values $> 0.05$ indicate that the model accounts well for the data. We also inspected the Watanabe–Akaike information criterion because it quantifies relative model fits by subtracting a correction for the number of parameters included. To test for differences in parameters between young and older participants, we subtracted the posterior

distributions of each parameter of the older group from the young group. We obtained posterior distributions of the age differences from which we extracted the mean and 95% credibility interval. An alternative approach to evaluating group differences consists in summing trial frequencies across young and older participants and fitting the aggregate data in a single extended MPT model. We conducted this analysis in order to further evaluate how the difference between $I_{down}$ and $I_{up}$ as well as between $S_{down}$ and $S_{up}$ changed across age groups (Supplementary Methods).

**Reporting summary**. Further information on research design is available in the Nature Portfolio Reporting Summary linked to this article.

## Results

A population of 52 participants (26 young and 26 older adults) took part in this study, with one young adult and six older adults being subsequently excluded (see Methods section for more details). Participants were screened for cognitive impairment to ensure they were within the age-related normative ranges for short-term, long-term, and spatial memory. Each participant completed a battery of visual tests, including visual acuity, contrast sensitivity, and kinetic

**Table 1 Summary of young and older participants' accuracy scores on the object recognition paradigm.**

| Group | Hit rate(M ± SD) | False alarm rate (M ± SD) | Correct rejection rate (M ± SD) |
|---|---|---|---|
| Young ($n = 25$) | 81.5% ± 14.6% | 4.4% ± 3.6% | 93.3% ± 5.1% |
| Older ($n = 20$) | 69.1% ± 17.3% | 10.2% ± 9.0 % | 82.1% ± 17.4% |

Hit rate, false alarm rate, and correct rejection rate are presented.
*M* mean, *SD* standard deviation.

perimetry, before performing a desktop-based source-monitoring task (Fig. 1). The experiment was divided into eight blocks, each comprising an encoding phase and a test phase. During the encoding phase, participants had to maintain fixation on a central cross while 30 objects were presented successively in the upper or lower part of the screen. During the test phase, 20 objects that had also been presented during the encoding phase and 10 new objects were shown one at a time in the center of the screen. Upon item presentation, participants had to decide if they had seen the object previously or not (i.e., item memory). If participants answered that they had seen the object, they then had to decide whether it had been presented in their upper or lower visual field (i.e., spatial memory).

Raw accuracy data for young and older participants are presented in Table 1. We analyzed participants' responses using standard linear *Pr* analysis and MPT modeling[54]. The latter can be understood in terms of binary branching trees (Fig. 2). Each branch represents a specific cognitive process and is associated with a parameter that measures the probability of its occurrence. Terminal nodes at the end of the branches represent the observed response categories. The parameters $I_{up}$ and $I_{down}$ relate to item memory, $S_{up}$ and $S_{down}$ relate to spatial memory, and the parameters o and g measure guessing rates (see Methods section for more details).

We first evaluated fits of the latent-trait hierarchical MPT model with posterior predictive *p*-values. The MPT model provided adequate fit to the data in terms of mean frequencies and covariance in both young adults ($p_{T1} = 0.51$, $p_{T2} = 0.44$) and older adults ($p_{T1} = 0.53$, $p_{T2} = 0.58$). We noted that the two parameters related to item memory ($I_{down}$ and $I_{up}$), the two parameters related to spatial memory ($S_{up}$ and $S_{down}$), and the two guessing parameters (g and o) were above 0 in young and older adults (Figs. 3, 4 & Supplementary Fig. 3). To examine differences in probabilities between young and older participants, we subtracted the posterior distributions of each parameter of the

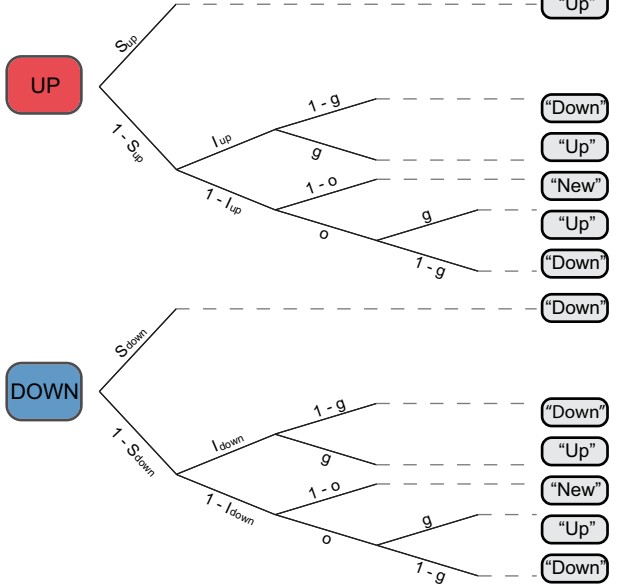

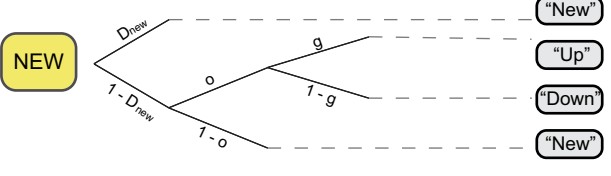

**Fig. 2 Graphical representation of the MPT *Source-Item* model used to analyze the experimental data from young and older participants.** The source-item multinomial processing tree model from the adapted source monitoring paradigm. A tree is used to model experimental data. Colored rounded rectangles represent the three different trial types: the object was presented in the upper part of the screen (UP), in the lower part of the screen (DOWN) or was not presented (NEW). Grey rounded rectangles represent participants' possible answers. $I_{up}$ and $I_{down}$ are the probabilities of remembering an item that was presented in the upper visual field and lower visual field, respectively. $S_{up}$ and $S_{down}$ are the probabilities of remembering the position of an item that was presented in the upper visual field and lower visual field, respectively. Parameter o refers to the probability of guessing that an item is old, while parameter g refers to the probability of guessing that an item was presented in the upper visual field.

## ITEM MEMORY

### (a) Standard *Pr* analysis

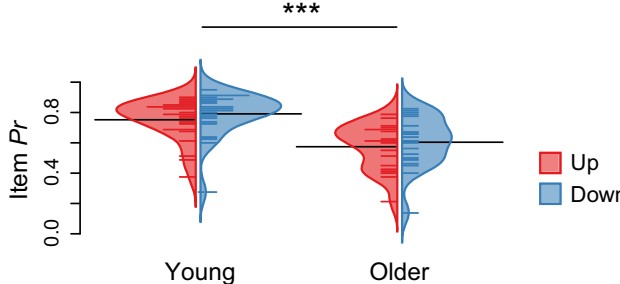

### (b) MPT analysis

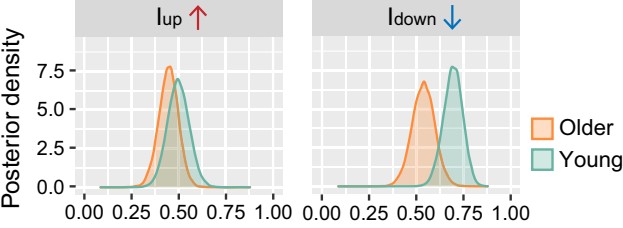

Mean group-level parameter estimates on probability scale

## SPATIAL MEMORY

### (a) Standard *Pr* analysis

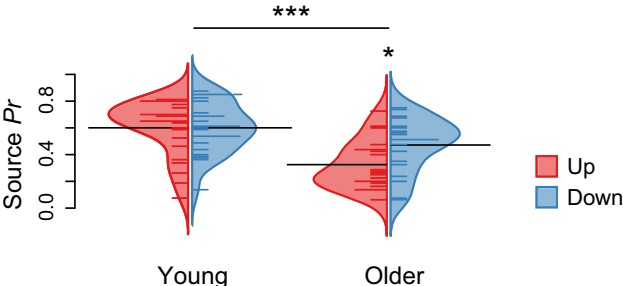

### (b) MPT analysis

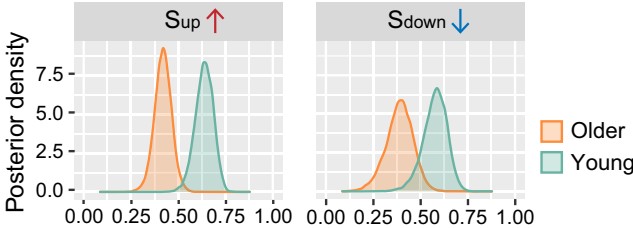

Mean group-level parameter estimates on probability scale

**Fig. 3 Behavioral performance on the item memory task evaluated with standard *Pr* and MPT analyses.** Item memory is tested during the object recognition task in which participants decide whether the object is old or new. **a** Bean plots depicting the results obtained using standard *Pr* analyses ($n = 25$ young + 20 older participants). Item *Pr* corresponds to the percentage of good answers minus the percentage of false alarms. We found evidence for main effects of age (F(1, 43) = 17.66, $p < 0.001$, $\eta_p^2 = 0.29$, 95% Confidence Interval (CI) [0.08, 0.48]) and object position (F(1, 44) = 11.58, $p = 0.0014$, $\eta_p^2 = 0.21$, 95% CI [0.04, 0.40]) on Item *Pr*. The bean plots provide the density curve for each group side-by-side, along with individual data points displayed in a colored rug-plot. Bold horizontal black lines correspond to the mean in each group. ***$p < 0.001$. **b** Posterior distributions depicting the results obtained using MPT analyses ($n = 25$ young + 20 older participants). The graphs show posterior distributions of the inverse-probit transformed group-level parameters related to item memory on the probability scale. The posterior distributions correspond to the updated knowledge about parameters I$_{up}$ and I$_{down}$ after considering the current data. I$_{up}$ is the probability of remembering an item that was presented in the upper part of the screen and I$_{down}$ is the probability of remembering an item that was presented in the lower part of the screen.

older group from the young group (Fig. 5). As a complementary analysis, we also tested whether item memory and source memory performance varied as a function of object position and age group in an extended MPT model with aggregate data (Supplementary Note 1, Supplementary Fig. 1).

**Relationship between item memory and objects' vertical position.** For the standard *Pr* analysis, we conducted a linear mixed model to investigate the effects of age group and object position on item *Pr*. The fit of the model was not improved by adding the interaction between age group and object position (with interaction: AIC = −150.91, without interaction: AIC = −152.69). The interaction was therefore not retained in subsequent analyses. We found a significant main effect of age group on item *Pr* (F(1, 43) = 17.66, $p < 0.001$, $\eta_p^2 = 0.29$, 95% CI [0.08, 0.48]). On average, item *Pr* was higher in

**Fig. 4 Behavioral performance on the spatial memory task evaluated with standard *Pr* and MPT analyses.** Spatial memory is tested during the spatial memory task in which participants decide whether the object appeared in the upper or lower visual field. **a** Bean plots depicting the results obtained using standard *Pr* analyses ($n = 25$ young + 20 older participants). Source *Pr* corresponds to the percentage of correct answers weighted by the number of answered trials. We found evidence for a main effect of age (F(1, 43) = 16.78, $p < 0.001$, $\eta_p^2 = 0.28$, 95% CI [0.08, 0.47]) and a significant interaction between age and object position (F(1, 43) = 5.37, $p = 0.025$, $\eta_p^2 = 0.11$, 95% CI [0.00, 0.30]) on Source *Pr*. The bean plots provide the density curve for each group side-by-side, along with individual data points in a colored rug-plot. Bold horizontal black lines correspond to the mean in each group. ***$p < 0.001$, *$p < 0.05$. **b** Posterior distributions depicting the results obtained using MPT analyses ($n = 25$ young + 20 older participants). The graphs show posterior distributions of the inverse-probit transformed group-level parameters related to spatial memory on the probability scale. The posterior distribution corresponds to updated knowledge about the parameters S$_{up}$ and S$_{down}$ after considering the current data. S$_{up}$ is the probability of remembering the position of an item that was presented in the upper part of the screen, and S$_{down}$ is the probability of remembering the position of an item that was presented in the lower part of the screen.

young adults (M = 0.77, SD = 0.14) than in older adults (M = 0.59, SD = 0.16) suggesting that young adults had better object recognition memory than older adults (Fig. 3a). We also observed a significant main effect of object position on item *Pr* (F(1, 44) = 11.58, $p = 0.0014$, $\eta_p^2 = 0.21$, 95% CI [0.04, 0.40]). Overall, participants were slightly better at recognizing objects that were presented in the lower visual field (M = 0.71, SD = 0.18) than in the upper visual field (M = 0.67, SD = 0.17). We conducted an additional linear mixed model that examined the influence of age group and object position on the distribution of misses. We found a main effect of age on the number of misses (F(1, 43) = 4.40, $p = 0.042$, $\eta_p^2 = 0.093$, 95% CI [0.00, 0.28]), indicating that older adults missed more item memory trials (M = 4.57, SD = 6.88) than did young adults (M = 1.87,

SD = 3.27). We did not reveal any evidence for an effect of object position (F(2, 88) = 1.40, p = 0.25, $\eta_p^2$ = 0.031, 95% CI [0.00, 0.12]). In other words, there was no statistically significant evidence of an unequal distribution of missed trials across up, down, and new objects.

Using a gamma generalized linear mixed model, we found a significant main effect of age ($\chi^2(1)$ = 24.32, p < 0.001, $\eta^2$ = 0.079, 95% CI = [0.068, 0.092]) on reaction times during the item memory task (Fig. 6a). This result reveals that older adults were slower (M = 1.28 s, SD = 0.46 s) to respond than young adults (M = 1.01 s, SD = 0.44 s). We also observed a main effect of object position ($\chi^2(1)$ = 5.69, p = 0.017, $\eta^2$ = 0.00044, 95% CI = [0.00030, 0.0014]): all participants were slightly slower in reacting to objects that had appeared in their upper visual field (M = 1.14 s, SD = 0.47 s) than in their lower visual field (M = 1.12 s, SD = 0.47 s). Finally, we showed that the block number had an impact on reaction times ($\chi^2(7)$ = 343.77, p < 0.001, $\eta^2$ = 0.030, 95% CI = [0.022, 0.040]), suggesting that

both young and older participants improved their performance throughout the experiment (Supplementary Fig. 2).

For the MPT analysis, we studied the two parameters related to item memory $I_{down}$ and $I_{up}$ (i.e., the probability of remembering an object presented in the lower and upper parts of the screen, respectively). We found no credible evidence that $I_{down}$ nor $I_{up}$ differed between age groups ($\Delta I_{down}$ = 0.156, 95% Bayesian confidence interval (BCI) [−0.001, 0.306]; $\Delta I_{up}$ = 0.052, 95% BCI [−0.099, 0.209]; Fig. 3b). Indeed, the parameters did not differ between young and older participants because their 95% BCI included 0. Therefore, there is no statistical evidence that the probability of having a sense of familiarity with an object, regardless of its position in space, differs between young and older adults. One may nonetheless note that the 95% BCI for $I_{down}$ fails to overlap 0 by a narrow margin. We found that older adults' probability of guessing that an item was presented during the encoding phase was significantly lower than young adults' ($\Delta o$ = −0.064, 95% BCI [−0.120, −0.020]; Supplementary Fig. 3). In other words, in the event that neither the object nor its position in space is remembered, young adults will answer that the item is old to a greater extent than will older adults.

**The vertical position of objects conditions spatial memory in healthy aging.** For the standard *Pr* analysis, we conducted a second linear mixed model analysis to investigate the effects of age group and item position on source *Pr*. Here, the fit of the model was improved by adding the interaction between age group and object position (with interaction: AIC = −39.08, without interaction: AIC = −35.79). The interaction was retained in subsequent analyses. We found a significant main effect of age group on source *Pr* (F(1, 43) = 16.78, p < 0.001, $\eta_p^2$ = 0.28, 95% CI [0.08, 0.47]). On average, source *Pr* was higher in young adults (M = 0.60, SD = 0.04) than in older adults (M = 0.33, SD = 0.04), highlighting better spatial memory performance in young adults compared with older adults (Fig. 4a). We also found the interaction between age group and object position to be significant (F(1, 43) = 5.37, p = 0.025, $\eta_p^2$ = 0.11, 95% CI [0.00, 0.30]). Older participants showed poorer performance (p = 0.017) when prompted on the position of items that were presented in the upper visual field (M = 0.33, SD = 0.18) compared with objects that were presented in the lower visual field (M = 0.47, SD = 0.20).

Notably, we verified that the latter result could not be attributable to the expected position of objects in space. Three independent raters assigned a pre-experimental source

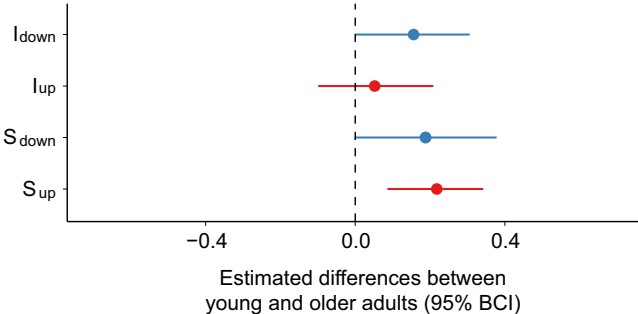

**Fig. 5 Differences in parameter estimates between young and older adults.** Forest plots showing the age difference in parameter estimates (Young minus Older) related to item and spatial memory (n = 25 young + 20 older participants). $I_{up}$ and $I_{down}$ are the probabilities of remembering an item that was presented in the upper visual field and lower visual field, respectively. $S_{up}$ and $S_{down}$ are the probabilities of remembering the position of an item that was presented in the upper visual field and lower visual field, respectively. Dots represent the mean parameter estimate difference and the horizontal lines are the 95% Bayesian credible interval (95% BCI) of this difference. Parameters in blue are those linked to the lower visual field while parameters in red are those linked to the upper visual field.

## (a) Item memory

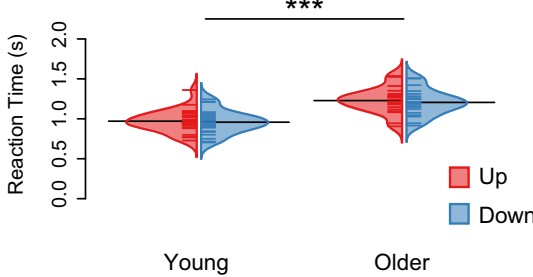

## (b) Spatial memory

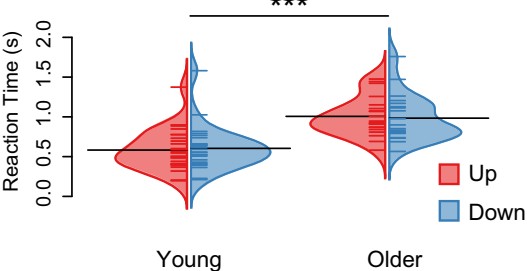

**Fig. 6 Reaction times in seconds across object positions and age groups.** Bean plots showing reaction times from the item memory and spatial memory tasks according to vertical object position and age (n = 25 young + 20 older participants). **a** We found evidence for main effects of age ($\chi^2(1)$ = 24.32, p < 0.001, $\eta^2$ = 0.079, 95% CI = [0.068, 0.092]) and object position ($\chi^2(1)$ = 5.69, p = 0.017, $\eta^2$ = 0.00044, 95% CI = [0.00030, 0.0014]) on reaction times during the item memory task. **b** We found evidence for a main effect of age ($\chi^2(1)$ = 21.83, p < 0.001, $\eta^2$ = 0.075, 95% CI = [0.062, 0.089]) but not of object position ($\chi^2(1)$ = 0.13, p = 0.72, $\eta^2$ = 0.00, 95% CI = [0.00, 0.00]) on reaction times during the spatial memory task. The bean plots provide the density curve for each object position side-by-side, along with individual data points displayed in a colored rug-plot. Bold horizontal black lines correspond to the mean of each group. ***p < 0.001.

**(a) Visual acuity**

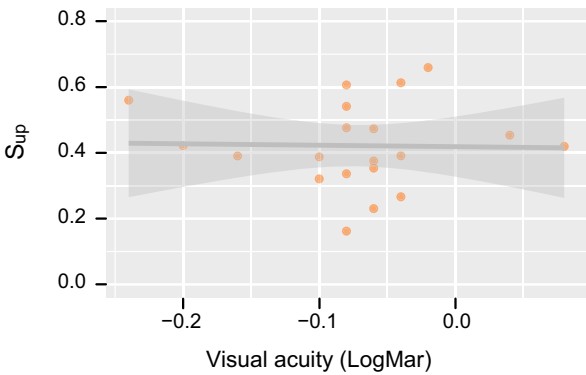

**(b) Contrast sensitivity**

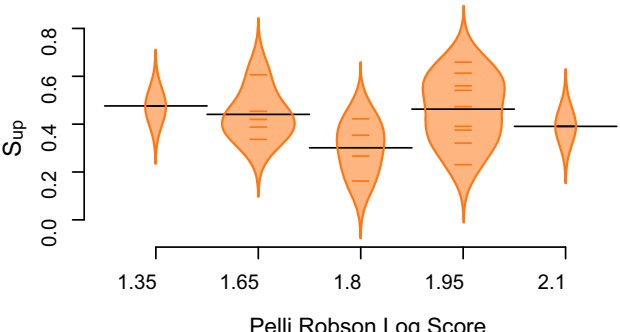

**Fig. 7 Associations between spatial memory performance and measures of visual function in older adults. a** Correlation plot depicting the probability of remembering the position of an object that was presented in the upper part of the screen ($S_{up}$) as a function of visual acuity assessed with the LogMAR scale in older adults ($n = 20$ older participants). We found no evidence for a significant association between $S_{up}$ and visual acuity in older participants ($R^2 = 0.011$, F(2, 17) = 0.099, $p = 0.91$, $\eta_p^2 = 0.002$, CI [0.00, 0.17]). Plot shows prediction line and confidence band of linear regression. **b** Bean plots showing the probability of remembering the position of an object that was presented in the upper part of the screen ($S_{up}$) across various Pelli-Robson contrast sensitivity scores in older adults ($n = 20$ older participants). We found no evidence for a significant association between $S_{up}$ and contrast sensitivity in older participants ($R^2 = 0.016$, F(2, 17) = 0.14, $p = 0.87$, $\eta_p^2 = 0.006$, CI [0.00, 0.22]). The bean plots provide the density curve for each log score separately, along with individual data points displayed in a colored rug-plot. Bold horizontal black lines correspond to the mean of each group.

expectation to each item included in the experiment (i.e., upper or lower expectation) and we kept the most frequent value. We conducted a linear mixed model to examine the effects of position and age on the percentage of correct source answers adjusting for the pre-experimental source expectation of each object. We included participants as random intercepts. We found no evidence for a significant influence of the pre-exposure (F(1, 1164) = 1.20, $p = 0.27$, $\eta_p^2 = 0.0010$, 95% CI [0.00, 0.008]), whereas the main effects of age (F(1, 45) = 14.14, $p < 0.001$, $\eta_p^2 = 0.24$, 95% CI [0.06, 0.43]) and position of item (F(1, 1164) = 10.08, $p = 0.0015$, $\eta_p^2 = 0.0086$, 95% CI [0.001, 0.02]) on spatial memory remained significant (Supplementary Fig. 4). Second, we also tested whether source decisions for false alarms were biased for the lower visual field. Indeed, a bias in source

location response for items that were never seen could imply that the objects were not well balanced in terms of pre-experimental source expectation. We computed the VMA for false alarms and conducted a linear regression to study the influence of age group on the VMA for false alarms (Supplementary Fig. 5). We found no statistically significant VMA difference between young and older adults ($R^2 = 0.021$, F(1,42) = 1.92, $p = 0.17$, $\eta_p^2 = 0.04$, 95% CI [0.00, 0.21]) suggesting that there is little evidence for a difference in the proportions of upper and lower false alarms across age groups. Moreover, the VMA was not significantly different from 0 as indicated by one-sided sign tests in young (M = 0.028, SD = 1.42, $p = 0.41$) and older participants (M = 0.55, SD = 1.01, $p = 0.06$). There is thus no statistical evidence for a bias in the choice of source location for false alarms.

Studying reaction times during the spatial memory task, we found a significant main effect of age on reaction times ($\chi^2(1) = 21.83$, $p < 0.001$, $\eta^2 = 0.075$, 95% CI = [0.062, 0.089]; Fig. 6b), meaning that older adults were slower (M = 1.26 s, SD = 1.16 s) to respond than young adults (M = 0.69 s, SD = 0.81 s). Despite the performance difference reported above, we revealed no evidence that object position affected reaction times during the spatial memory task ($\chi^2(1) = 0.13$, $p = 0.72$, $\eta^2 = 0.00$, 95% CI = [0.00, 0.00]). We also showed a significant main effect of block number ($\chi^2(7) = 867.37$, $p < 0.001$, $\eta^2 = 0.046$, 95% CI = [0.035, 0.056]) as well as a significant effect of the interaction between age group and block number on reaction times ($\chi^2(7) = 39.18$, $p < 0.001$, $\eta^2 = 0.013$, 95% CI = [0.0070, 0.019]). Although all participants improved their reaction times on the spatial memory task across blocks, older adults showed a steeper learning curve (*first block*: M = 2.06 s, SD = 1.58 s; *last block*: M = 0.95 s, SD = 0.88 s) than young adults (*first block*: M = 0.99 s, SD = 0.93 s; *last block*: M = 0.60 s, SD = 0.80 s; Supplementary Fig. 2).

For the MPT analysis, we next studied the variables related to spatial memory $S_{down}$ and $S_{up}$ (i.e., the probability of remembering the position of an object presented in the lower and upper part of the screen, respectively). We found that while $S_{down}$ was equivalent between age groups ($\Delta S_{down} = 0.188$, 95% BCI [−0.002, 0.378]), $S_{up}$ differed significantly between young and older adults ($\Delta S_{up} = 0.218$, 95% BCI [0.086, 0.342]; Fig. 4b). These results suggest that older adults were less likely to recall the position of objects presented in the upper visual field. Worthy of note, the guessing parameter g, defined as the probability of guessing that an item was presented in the upper visual field, was equivalent in young and older participants ($\Delta g = 0.080$, 95% BCI [−0.079, 0.239]; Supplementary Fig. 3).

**Measures of visual function do not explain the age-related upper visual field deficit.** We sought to verify whether the influence of the vertical position of information on spatial memory in older adults could emerge from variations in visual acuity or contrast sensitivity. Reports have shown that these visual function measures are not equivalent across the upper and lower visual fields[9,60]. Considering the latter, we conducted linear regression analyses to explore the effects of visual acuity and contrast sensitivity on older adults' probability of remembering the position of an object presented in the upper part of the screen ($S_{up}$; Fig. 7). We found little evidence for a significant influence of visual acuity ($R^2 = 0.011$, F(2, 17) = 0.099, $p = 0.91$, $\eta_p^2 = 0.002$, 95% CI [0.00, 0.17]) or contrast sensitivity ($R^2 = 0.016$, F(2, 17) = 0.14, $p = 0.87$, $\eta_p^2 = 0.006$, 95% CI [0.00, 0.22]) on the age-related upper visual field deficit. Next, we wondered whether the observed upper visual field deficit in older adults could arise from differences in the ratio of upper to lower visual field area as assessed by kinetic perimetry.

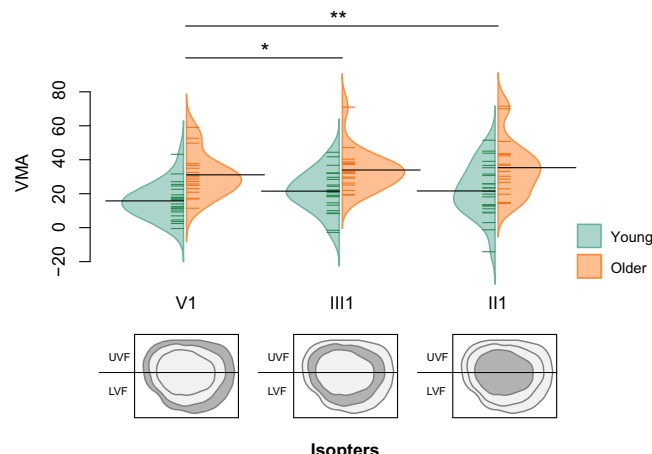

**Fig. 8 VMA for the visual field areas across three isopters in young and older adults.** Bean plots depicting the vertical meridian asymmetry (VMA) for visual field areas across isopters V1, III1, and II1 ($n = 25$ young + 18 older adults). Trained orthoptists measured visual field areas with kinetic perimetry for each participant. The vertical meridian asymmetry computes the discrepancy between the lower and upper visual field areas. A score of 0 indicates no difference between the upper and lower visual field areas. A score superior to 0 indicates a greater area for the lower visual field and a score inferior to 0 indicates a greater area for the upper visual field. We found evidence for main effects of age ($F(1, 40) = 6.75$, $p = 0.013$, $\eta_p^2 = 0.14$, CI [0.007, 0.35]) and isopter ($F(2, 213) = 5.90$, $p = 0.0032$, $\eta_p^2 = 0.05$, CI [0.007, 0.12]) on VMA for visual field areas. $**p < 0.01$, $*p < 0.05$. The bean plots provide the density curve for each age group side-by-side, along with individual data points displayed in a colored rug-plot. Bold horizontal black lines correspond to the mean of each group. The studied isopter is colored in dark gray below the beanplot. UVF upper visual field, LVF lower visual field.

We conducted linear mixed models to study the effects of age group, $S_{up}$, and isopter on the VMA. The interaction between $S_{up}$ and age group did not improve the model significantly (with interaction: AIC = 2044.4; without interaction: AIC = 2043.5), and it was therefore not retained in subsequent analyses. As expected, we first found that age group had a significant impact on the VMA ($F(1, 40) = 6.75$, $p = 0.013$, $\eta_p^2 = 0.14$, 95% CI [0.007, 0.35]; Fig. 8), with older adults having significantly higher VMAs (M = 33.47, SD = 16.04) than young adults (M = 19.63, SD = 13.55). This result suggests that the aging process affects the upper visual field area more than the lower visual field area. There was also a main effect of isopter on the VMA ($F(2, 213) = 5.90$, $p = 0.0032$, $\eta_p^2 = 0.05$, 95% CI [0.007, 0.12]), with significant differences between V1 and III1 ($t(213) = -4.53$, $p = 0.017$, d = 0.32, SE = 0.153) and V1 and II1 ($t(213) = -5.17$, $p = 0.0053$, d = 0.31, SE = 0.153). Finally, we revealed no significant effect of $S_{up}$ on the VMA ($F(1, 40) = 2.26$, $p = 0.14$, $\eta_p^2 = 0.05$, 95% CI [0.00, 0.23]). There is thus no evidence for an association between the more pronounced VMA in older participants and the reported performance asymmetries in spatial memory.

## Discussion

The aim of this study was to test the hypothesis that the visual encoding of object attributes plays an essential role in maintaining spatial cognition across the lifespan. More specifically, we examined whether the vertical position of visual information affects item and spatial memory, and whether it does so differentially in young and healthy older adults. The findings revealed a general lower field advantage for item recognition and an age-related decline in spatial memory for objects presented in the upper visual field only. The latter result was not found to be explained by participants' visual field area asymmetries, suggesting that this specific spatial memory deficit may originate from higher visual regions.

**Familiarity-based item memory is preserved in healthy aging.** While we observed that spatial memory was consistently impaired in older adults, the results pertaining to item memory did not converge across analysis methods. It is well documented that healthy aging is accompanied by extensive changes in object–location binding that are usually more pronounced than changes in simple item recognition[61–65]. The prominent associative memory theory claims that age-related deficits in episodic memory emanate primarily from inadequate associations between items and contextual information[63,66–68]. A long line of research asserts that episodic recollection is impaired whereas familiarity and gist processing are aspects of item memory that remain intact in older age[63,69,70]. The source-item model of memory used to explain the present data defines item memory as the probability of remembering an item given that spatial position has not been recalled. One can argue that such a process is more akin to familiarity (i.e., a global measure of memory strength) than to a precise recollection of the object. The Item $Pr$ measure of accuracy from the standard analyses, on the other hand, could reflect a recollection mechanism that involves a more detailed retrieval of past information. Indeed, previous research has highlighted the impossibility for item and spatial memory to be precisely disentangled using traditional statistical methods[54]. Our study, therefore, confirms that the general memory of an object is preserved and that its detailed recollection is impacted in healthy aging.

**Spatial memory is impacted in the upper visual field of older adults.** An effect of the objects' vertical position on item memory was found across all participants. Object recognition and reaction times were improved for items presented in the lower portion of the screen. In the literature, object processing tasks including letter identification, target localization, word discrimination, and face sex-categorization, have been conferred an upper visual field advantage[25,26,71,72]. One possibility is that visual experience with specific stimuli shapes the formation of vertical visual field biases. For example, researchers are evoking the possibility that faces are better encoded in the upper visual field because they are situated on the upper portion of a body[31,73]. That everyday objects are more frequently encountered in the lower visual field may explain why the encoding of items in our task was facilitated in the lower portion of the screen for all participants. Regarding spatial memory, our study provides nuance to the widespread view that older adults are impaired at recalling spatial contextual information. We revealed that older adults have worse spatial memory than young adults but only for objects presented in the upper visual field. In other words, spatial memory may be preserved in aging for visual information situated in the lower field. Of note, while the expectation of the position of objects in space may contribute in part to the lower visual field advantage for item recognition, we demonstrated that the spatial memory deficit in older adults was most likely attributable to an altogether different mechanism. Two previous studies reported an age-related loss of function in the upper visual field for rapid stimuli detection[34,35]. We extended their work to high-level cognitive functions and showed that in older age, the mnemonic trace of an object is dependent upon its vertical position in space.

Importantly, the position of information interacted with age for spatial memory performance but not for item recognition. Such a result strongly implies that the observed upper visual field

deficit in spatial memory does not arise from a biased distribution of covert attentional processes, or the same deficit would have emerged for item memory. In agreement with this conclusion, accumulating evidence postulates that performance asymmetries across the vertical visual fields are not a product of attentional differences[8,11,74]. Moreover, the fact that object location influences item recognition equivalently in young and older adults speaks to the spatial nature of the upper field decline in aging. Age-related differences in object processing in the upper visual field appeared only when the spatial properties of objects were solicited. Along the same lines, a study concluded that upper–lower performance asymmetries were primarily related to the visuospatial features of a stimulus[28].

**The biological bases of vertical asymmetries in mnemonic performance.** The underpinning biological bases for the reported lower visual field benefit in item recognition and the upper visual field spatial memory deficit in older adults are unknown. They could be located anywhere along the visual pathway, from the retina to higher neural structures. A recent computational model highlighted that the VMA for contrast sensitivity was only weakly attributable to retinal factors including optics and cone sampling[75]. Furthermore, we excluded the possibility that low-level perceptual differences hindered the ability of older adults to create a detailed mnemonic trace of the object and its position. Indeed, we showed that neither visual acuity, contrast sensitivity, nor the VMA for visual field areas was associated with the probability of remembering the position of an object that was presented in the upper visual field in older participants. The VMA for visual field areas reflects the sensitivity threshold to dots of various sizes and light intensities but it doesn't provide information relative to how that signal is processed. Notably, cortical inhomogeneities in V1 have been shown to covary with spatial frequency and contrast sensitivity behavioral asymmetries across the vertical visual field[60,76]. It is therefore likely that top-down influences from higher visual regions play a role in the general lower field preference for item recognition and in the age-related spatial memory decline specific to the upper visual field. Visual information is propagated from early visual areas through the ventral and dorsal streams that bear distinct proportions of upper and lower visual field afferents[77]. The ventral stream benefits from a larger upper visual field representation and it may be involved in far-space object identification. In contrast, the dorsal stream relies more strongly on lower visual field inputs and could play a role in near-space visuomotor processing[78]. The slight lower field benefit for item memory in young and older adults is in line with fMRI studies reporting that the lateral occipital visual area, an object-selective region, has stronger BOLD responses to objects in the lower than in the upper visual field[79–81]. Importantly, regions upward of the ventral stream are closely associated with pattern separation, an essential ability to reduce the overlap between similar object locations[82]. Therefore, one possibility for the upper visual field deficit in spatial memory is an age-related neural loss specific to regions upward of the ventral stream. This study lends credence to the idea that the visual field biases contained in higher visual regions could have specific behavioral correlates in older adult populations.

**Limitations.** This study has limitations pertaining to the choice of stimuli and to the task design. We presented items that were devoid of any context on a blank screen. The present experiment should be replicated using natural scenes with objects that carry realistic contextual information. Furthermore, the items chosen are everyday objects that are more often encountered in the lower visual field. Even though we found no statistical evidence for a pre-experimental source expectation bias, it would be pertinent for future research to include more objects that are associated with the upper visual field. Regarding task design, we highlight that the results should be confirmed with less cognitively demanding mnemonic tasks. Indeed, the paradigm required participants to encode thirty items and their position before being tested. Finally, our results apply only to non-fixated objects, and we stress that an important next step in the field is to investigate whether free-viewing conditions can rescue older adults' spatial memory deficit in the upper visual field.

## Conclusion

Our main finding that the vertical position of objects determines spatial memory performance in healthy aging strongly conveys the importance of object feature encoding in the context of cognitive decline across the lifespan. The risk of falling, the stooped posture, and the careful avoidance of obstacles on the ground are only a few examples of physical and perceptual modifications in aging that could reshape older adults' use of visual space. Our findings also have far-reaching implications for complex cognitive skills that require adequate memory of the location of objects, such as spatial navigation or driving. These abilities require the processing and spatial encoding of visual information that is most often found in the upper visual field: large immovable objects, road signs, monuments, or buildings. In such contexts, we argue that older adults could benefit from using cues situated in the lower portion of their visual field or from actively training their upper visual field. Moreover, to maintain older adults' autonomy and mobility, age-friendly environmental adaptations should be made in line with the vertical visual field preferences hereby documented. Finally, future research efforts should be committed to uncovering the neurobiological factors that link age-related visual, behavioral, and cognitive impairments.

## Data availability
The experimental data that support the findings of this study are available via the OSF repository at: https://osf.io/unby4/ (https://doi.org/10.17605/OSF.IO/UNBY4). Inside each subsection of the repository (e.g., *Visual Fields (data and script)*) is a text file that lists the figures created with the source data available in that specific folder.

## Code availability
Custom codes used to run the standard *Pr* and MPT analyses and to plot the results are available at: https://osf.io/unby4/[83]. The codes were written using R version 4.0.3 in RStudio version 1.4.1103 (R Core Team, 2020; RStudio Team, 2021).

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

## Acknowledgements

The authors would like to express their most sincere gratitude to the volunteers who took part in this study. We also thank Elisa Tartaglia and Valérie Parmentier (Essilor International) for their helpful inputs. This research was supported by the Fondation pour la Recherche sur Alzheimer, the French National Research Agency (ANR-18-CHIN-0002), the LabEx LIFESENSES (ANR-10-LABX-65), and the IHU FOReSIGHT (ANR-18-IAHU-01). The funders had no role in study design, data collection and analysis, decision to publish or preparation of the manuscript.

## Author contributions

Study design: M.D., S.R., A.A.; Data acquisition: M.D., L.V.P., S.C.; Data processing: M.D., B.B.; Manuscript writing: M.D., S.R., J.A.S., A.A.

## Competing interests

The authors declare no competing interests.
