## [Peer Review File · Communications Psychology]

5th Jan 23

Dear Dr Arleo,

Thank you for your patience during the peer-review process. Your manuscript titled "The Vertical Position of Visual Information Conditions Spatial Memory Performance in Healthy Aging" has now been seen by 3 reviewers, whose comments are appended below. I have discussed the reports with my colleagues and I regret to inform you that we decided that in light of the referee reports, we cannot publish your manuscript in Communications Psychology.

The reviewers express enthusiasm about the research question and the potential novel insights. However, you will see that they also raise substantive concerns, in particular with regard to the strength evidence put forward to support your conclusions. Taking these points together with our editorial considerations, these reservations preclude publication of this study in Communications Psychology.

In particular, Reviewer #1 raises concerns about potential confounds, which would require additional experimental work (new data). Similarly, the concerns voiced by Reviewer #2 would also best be addressed through additional data collection, while issues highlighted by Reviewer #3 (and also Reviewer #2) would necessitate a significant reanalysis of the existing and new data. Addressing these concerns presents an amount of work that extends beyond what is considered an appropriate editorial request of an extended revision.

I am sorry that we cannot be more positive on this occasion and thank you for the opportunity to consider your work.

Best regards,

Jennifer Bellingtier

Jennifer Bellingtier, PhD
Senior Editor
Communications Psychology

REVIEWERS' EXPERTISE:

Reviewer #1 visual attention, memory

Reviewer #2 visual attention, memory

Reviewer #3 memory, computational analysis

REVIEWER COMMENTS:

Reviewer #1 (Remarks to the Author):

I reviewed the research report manuscript titled: "The Vertical Position of Visual Information Conditions Spatial Memory Performance in Healthy Aging".

This study evaluates the impact of the vertical localization of items on source memory in aging, and essentially shows a significantly lower accuracy of source memory in older vs. younger adults for items encoded in the upper field, but not for items encoded in the lower field. This finding holds from two relevant and independent types of data analyses, namely a multinomial processing tree (MPT) modeling approach and a more classical linear approach based on estimating discrimination indices. To probe whether this main finding was merely emerging from visual impairment, the authors further tested whether the source memory accuracy on the upper-field-encoded items was related to visual acuity, contrast sensitivity or kinetic perimetry. They found no evidence for that. Item memory as well as reaction times were also investigated, and presented an impact of age, but not of the encoding visual field. In general, this study was motivated by the idea of extending the exploration of the 'vertical meridian asymmetry' (VMA) phenomenon, by which numerous visual abilities differ between the upper and lower visual fields, to higher cognitive functions like memory, or, more precisely here, to source memory. Importantly, this was also addressed regarding the impact of aging.

In light of their findings, the authors argue that the age-related VMA observed here for spatial memory disfavoring the upper field is in fact independent from the previously shown VMA pertaining to vision, appears with age contrarily to visual VMA, and might interestingly originate from higher visual regions.

Overall, I found the manuscript very well written and justified, both theoretically and methodologically. The conclusion is quite original and has indeed the potential to be of interest for the community and the wider field. However, I believe that one important dimension of the experiment (the 'pre-experimental source expectation', see major concern) still remains to be addressed for the conclusion to be convincing. On a less important note, I also think that the question of the ecological importance of memory for non-directly-gazed items needs to be discussed more for the overall impact to be clear (see last minor comment).

Major concern:

- There is evidence that objects that are usually in further distances are usually seen in the upper field, and that those usually at close distances are usually seen in the lower field (see for instance Pevic, 1990, cited in the manuscript). Based on that, it is reasonable to expect that the judgment of the vertical location where an item was at encoding will be biased towards its usually seen vertical field. This 'pre-experimental source expectation' is key in the scope of the goals and claims of this study, but is however not explicitly controlled, nor discussed in the manuscript. I propose below a series of points to discuss, and analyzes to run to address the consequent current uncertainty of the main claim.

While eliminating objects "strongly associated with the ground, the ceiling, or the sky" from the stimulus sets and counterbalancing items across participants as it was done here are already important careful controls, I would argue that these might not be sufficient in avoiding any bias relative to the overall pre-experimental source expectation. Specifically, based on pre-experimental exposure in daily life, the objects chosen could for instance tend to be overall expected in the lower visual field. If this is the case, then the overall source memory responses might have been biased towards the response 'down', and thereby the accuracy towards being better on the lower-field-encoded items and worse on the upper-field-encoded items. Since it is expected that this bias would operate more in poorer memory situations and known that older adults are impaired on episodic memory such as source

memory, a bias in the items pre-experimental source expectation could explain the main pattern of results, compromising the current claimed interpretation. Ideally, it would have been interesting to separate the items in two controlled and balanced groups (one with pre-experimental source expectation in the upper visual field, and the other in the lower visual field) to evenly distribute the items in each condition and/or assess their putative impact. Without this, it could still be interesting to assess this vertical visual field of the pre-experimental source expectation of each item used, and take this variable into account in the analyses (e.g. as a trial-level covariate in the analyses, or otherwise at the condition level). At least, it would be important here to report whether a bias in the source location reported exists on false alarms (responses “old” on ‘new’ items), i.e. whether the parameter ‘g’ (in the MPT modeling approach) and/or whether the number of source location of type ‘down’ reported on false alarms (in a more classical approach) are/is different than the chance level. Showing such bias on items never seen in the upper or lower visual field would disfavor the interpretation of an actual VMA for source memory (in line with its lack of relationship with participants visual VMA). On the contrary, the absence of such bias would suggest that the stimuli were well balanced in terms of pre-experimental source expectation, making the central claim of the manuscript stronger.

Minor comments:

- Considering how the parameters ‘o’ and ‘g’ are defined in the MPT modeling approach (second paragraph of Results section, and in Figure 2), my understanding is that they don't correspond to the “probability of correctly guessing that an item was presented [in the upper visual field, for g]” as it is written along the manuscript, but instead to the “probability of guessing that an item was presented [in the upper visual field, for g]”. Indeed, whatever the item (target or distractor), ‘o’ corresponds to the probability of guessing that the item is ‘old’ and not ‘new’, and ‘g’ to the probability of guessing that the item was presented in the upper (‘up’) and not lower (‘down’) visual field, such that, even if responding ‘old’ or ‘up’ would be incorrect on distractors, this would be more probable with ‘o’ or ‘g’ being higher. In the scope of being even more in line with how the parameters in the MPT modeling approach here are defined (e.g. second paragraph of the Results section), since each parameter measures the “probability of occurrence” - rather than the “accuracy” - of a specific cognitive process, this also applies to the specific cognitive processes of ‘guessing old’ (‘o’) and ‘guessing up’ (‘g’). This should be corrected along the manuscript (specifically in the Results and Methods sections).
- It would be interesting, and more consistent with other graphs of the manuscript, to plot figures 2B and 2C in the same format than the others (e.g. figure 3), in particular to have the same visualization of the subject values distribution.
- While the topic tackled in this study is interesting and promising for better understanding the cognitive impairment of aging and accompanying the aging population with age-friendly strategies like the authors underlined, it must be acknowledged that it only covers the case of source memory when the actual items are encoded in the absence of direct saccade on these. In real life though, it might be more ecologically the case that we actually look at the items - and thereby saccade to them - to encode them. Moreover, albeit of smaller importance, these results on non-directly-gazed items only apply to source memory and not item memory, meaning that they are more concerning situations of remembering the vertical location where an item - only seen in the periphery - was situated. I believe these aspects

relative to the ecological importance of the results should be more explicitly discussed, and the impact of this result in daily life strategy mitigated and/or discussed accordingly.

Reviewer #2 (Remarks to the Author):

The authors conducted an interesting study of the influence of target location in upper vs, lower visual field on subsequent item recognition memory and “source memory” for that item’s location. The underlying issue – whether more primary deficits in visual processing might be responsible for age-differences in episodic memory – is an important one, but I have some concerns about the study, which I hope the authors might address.

Introduction: The authors provide an interesting list of tasks in which there are upper vs. lower visual field advantages. Perhaps the authors could comment on whether the distinction between tasks with upper and lower visual field advantages is related to whether attentional functions are more implicated (e.g., visual search, change detection) or less implicated (e.g., hue discrimination, contrast sensitivity). That might be relevant to the two mnemonic abilities the authors explore.

Methods: Did the eye tracking data indicate how well participants were actually able to maintain fixation on the central cross during the lengthy (2500 ms) presentation of the stimuli? Were there the same degree of “improper” saccades to the object pictures for upper and lower offsets?

Why were participants only tested on 20 studied items (along with 10 foils) out of the 30 that they had studied in each run?

Results: While the MPT analysis employed has its advantages, presenting the results only in terms of the parameters of the model is opaque to any readers who are not familiar with the method. The authors should more fully explain the exact meaning of what they are presenting in the figures.

Additionally, before presenting the results in terms of the MPT models, the authors should present the raw accuracy data.

While the statistics indicate non-significant differences between the parameter estimates of younger and older adults for I-down and S-down conditions, with some overlap in the distributions, it does not seem reasonable to categorically state that the groups do not differ, relative to the clearer overlap apparent in I-up. I would like to see the results of standard simple analyses of d' for items and location remembering for correct endorsed items. Despite its disadvantages, those simpler analyses do not make inherent assumptions regarding dual-process or single-process bases of the data, and it seems to me important to present those analyses along with the MPT parameter analyses.

Another aspect of the data that is not transparent in the reported parameters is whether those parameters reflect the discrimination accuracy in each visual field, or the degree of endorsement of stimuli presented in that VF? In other words, does it reflect something like hit rate, or something like d' ? I guess I’m wondering whether older adults had poorer performance in UVF because of a bias towards responding “lower”. I’m a little confused here, because spatial memory ability is confounded with the location of the stimulus presentation. Had the authors presented objects in 4 quadrants, and then asked whether the target appeared on the right or the left, and found that discrimination was different for older and younger adults only in the upper VF, that would have been a clearer indication of different

abilities for upper and lower VF presentations. But since in the actual paradigm any miss from UVF is a false alarm for LVF, how can that be unentangled to say that memory was better in LVF?

I also find some of the reported results non-intuitive, for example, the parameter estimate indicating that “older adults’ probability of correctly guessing that an item was presented during the encoding phase was significantly lower than young adults” – does that mean that they guessed to the same degree, but were less successful? Or that they simply guessed less? Since this is a contention based on a parameter extracted from the model, it might not correspond to actual cognitive processes. Perhaps the authors can explain how this emerged from their models.

Reviewer #3 (Remarks to the Author):

The authors use a MPT approach to compare spatial and item memory performance in younger and older adults as a function of whether stimuli were presented in the upper or lower hemifield during study. Using a Source-Item model introduced by Cooper et al. (2017), the authors conclude that item memory is invariant across upper and lower positions for both young and old participants, but that spatial memory is impaired in older participants for list items presented in the upper visual field.

The study tackles an interesting question using a principled theoretical analysis. The specific Source-Item model used here has been criticized by Kellen and Singmann (2017) on the grounds of parameter polysemy, which is troubling. This means that common parameters appearing in different processing trees (e.g., Up/Down vs. New, from Figure 2) are defined slightly differently and reflect slightly different psychological processes.

Further, the analyses are conducted in a way that leaves a lot of information unexamined. This information, I believe, leads to an interpretation of the data that is potentially quite different from what the authors conclude based on the current analyses. I am therefore concerned that the conclusions do not necessarily follow from the data. Minimally, there are some potential caveats and qualifications that might need to be made. I detail my concerns below.

Major Concerns

1 - Given the importance of good vision in the task, I was surprised that participants requiring glasses were not screened out. What proportion of younger/older participants who normally wear glasses completed the task without them? If these participants are unevenly distributed, or the visual deficit corrected by glasses is larger in one group vs. another, might this explain an apparent spatial memory deficit (e.g., due to greater variability in encoding location information)?

2 - I have concerns about the way the MPT has been specified. For one, the model is not described in sufficient detail. For example, parameters o and g are described with respect to old items, but not with respect to new items. This is consequential because the parameters appear in both old and new decision trees, but their meaning differs across them. This issue was discussed by Kellen and Singmann (2017) in a comment to the Cooper et al. (2017) article the authors cite. On the one hand, the description of the o and g parameters are

inaccurate in the text—they do not describe the probabilities of correctly guessing “old” or “up” (for stimuli appearing in the upper position), as stated. If this were true, these parameters could not appear in the “New” decision tree because these branches all lead to errors. More fundamental is that the o parameter in the “New” tree refers to the probability of falsely recognizing a new item as old, conditioned on the failure to recollect combined item and source information about it. However, in the two “Old” trees, the o parameter refers to the probability of correctly identifying an old item as old conditioned on both the failure of recollection and the subsequent failure of familiarity-based recognizing. That the parameters perform different duties depending on the stimulus makes them difficult to interpret because they are not characterizing the same psychological process.

Kellen, D. & Singmann, H. (2017). Memory representations, tree structures, and parameter polysemy: Comment on Cooper, Greve, and Henson (2017), *Cortex*, 96, 148-155.
<http://dx.doi.org/10.1016/j.cortex.2017.05.015>

3 - It is unclear how the data were modelled using the MPT. Were the individual participant data modelled, and the resulting posterior estimates compared across groups, or were the data modelled hierarchically? The latter constitutes best practice, but either way, some clarification is needed.

4 - The comparisons of the I_Up and I_Down parameters don't tell the whole story. While performance is similar in the lower position, the evidence for no difference in the upper position is less convincing—credible intervals fail to overlap 0 by the narrowest of margins. Perhaps model selection might provide a better test—does a model with parameters constrained to be equal across groups outperform the one presented by the authors? My sense, however, is that the question of importance is whether younger and older participants show similar patterns of performance across the two positions rather than absolute levels of performance within each position. That is, the difference between I_Up and I_Down within each participant group would be a more meaningful comparison to make across groups (viz. are the asymmetries comparable across age groups). A model selection way of formulating the question would set the parameter differences to be equal across groups. The critical comparison is whether this constrained model is outperformed by one where different parameter differences are required for older and younger groups. (Similar comments apply to the comparisons of S_Up and S_Down .)

5 - The results are mainly interpreted through the lens of a relative performance deficit for spatial information in the upper visual hemifield. The general thrust of the manuscript seems at odds with the visual trends in the data shown in Figure 2. Numerically, at least, it would appear that both groups exhibit a familiarity-based item recognition deficit in the upper vs. lower field and a spatial recollection benefit for the upper field. Given the issues with the analyses discussed above, I am not confident that the conclusions are appropriately supported by the data.

Minor Concerns

1 - Lines 287-288 – Presentation time for items should be specified as 2500 ms per item. Current wording could be taken to imply that 30 objects were presented in 2500 ms.

2 - Line 293 – Can you specify the nature of the pseudorandomization of old and new items at test?

3 - Lines 322-323 – “We excluded two older subjects from the visual field analyses because their kinetic perimetry data were not exploitable.” I was unsure what was meant by “exploitable” here.

I suggest that you consider Scientific Reports as a suitable venue for your work. To transfer your manuscript there, please use our [link redacted] manuscript transfer porta. You will not have to re-supply manuscript metadata and files, unless you wish to make modifications, but please note that this link can only be used once and remains active until used. For more information, please see our [link redacted] page.

Note that any decision to opt in to In Review at the original journal is not sent to the receiving journal on transfer. You can opt in to *[In Review](https://www.nature.com/nature-portfolio/for-authors/in-review)* at receiving journals that support this service by choosing to modify your manuscript on transfer. In Review is available for primary research manuscript types only.

Note on appeals: In exceptional circumstances, it is in authors’ interest to appeal an editorial decision. More information on appeals is available here: <https://www.nature.com/commpsychol/submit/resources#appeals>

17th Jan 23

Dear Dr Arleo,

Thank you for your correspondence asking us to reconsider our decision on your Article, "The Vertical Position of Visual Information Conditions Spatial Memory Performance in Healthy Aging". After careful consideration we have decided that we would be willing to consider a revised version of your manuscript.

Along with your revised manuscript, you should also submit a separate point-by-point response to all of the concerns raised by the referees, in each case describing what changes have been made to the manuscript or, alternatively, if no action has been taken, providing a compelling argument for why that is the case. If we feel that a substantial attempt has been made to address the editorial concerns and referees' comments, this response will be sent back to the referees - along with the revised manuscript - so that they can judge whether their concerns have been addressed satisfactorily or otherwise.

I should stress, however, that we would be reluctant to trouble our referees again unless we thought that their comments had been addressed in full.

When revising your paper please:

- ensure that it complies with the editorial policies outlined below
- ensure it meets our format requirements as set out in our [Guide to Authors](http://www.nature.com/nathumbehav/info/gta) and highlighted below.
- ensure that the statistics reporting and interpretation is in line with journal guidelines <https://www.nature.com/commspsychol/submit/submission-guidelines#statistical-guidelines>

Please mark all correspondence via email with your Communications Psychology reference number in the subject line.

If the revision process takes significantly longer than five months, we will be happy to reconsider your paper at a later date, provided it still presents a significant contribution to the literature at that stage.

Please use the following link to submit your revised manuscript, point-by-point response to the Reviewers' comments with a list of your changes to the manuscript text (which should be in a separate document to any cover letter) and any completed checklist:

[link redacted]

Best regards,

Jennifer Bellingtier

Jennifer Bellingtier, PhD
Senior Editor
Communications Psychology

EDITORIAL POLICIES AND FORMATTING

Editorial Policy: [Policy requirements](https://www.nature.com/documents/nr-editorial-policy-checklist.pdf) (Download the link to your computer as a PDF.)

Furthermore, please align your manuscript with our format requirements, which are summarized on the following checklist:

[Communications Psychology formatting checklist](https://www.nature.com/documents/commsj-psychol-style-formatting-checklist-article.pdf)

and also in our style and formatting guide [Communications Psychology formatting guide](https://www.nature.com/documents/commsj-psychol-style-formatting-guide-accept.pdf) .

* **CODE AVAILABILITY:** All Communications Psychology manuscripts must include a section titled "Code Availability" at the end of the methods section. In the event of publication, we require that the custom analysis code supporting your conclusions is made available in a publicly accessible repository; please choose a repository that provides a DOI for the code; the link to the repository and the DOI must be included in the Code Availability statement. Publication as Supplementary Information will not suffice. We ask you to prepare and upload code at this stage, to avoid delays later on in the process.

* **DATA AVAILABILITY:**

All Communications Psychology research manuscripts must include a section titled "Data Availability" at the end of the Methods section or main text (if no Methods). More information on this policy, is available at <http://www.nature.com/authors/policies/data/data-availability-statements-data-citations.pdf>.

At a minimum the Data availability statement must explain how the data can be obtained and whether there are any restrictions on data sharing. Communications Psychology strongly endorses open sharing of data. If you do make your data openly available, please include in the statement:

We recommend submitting the data to discipline-specific, community-recognized repositories, where possible and a list of recommended repositories is provided at <http://www.nature.com/sdata/policies/repositories>.

If a community resource is unavailable, data can be submitted to generalist repositories such as [figshare](https://figshare.com/) or [Dryad Digital Repository](http://datadryad.org/). Please provide a unique identifier for the data (for example a DOI or a permanent URL) in the data availability statement, if possible. If the repository does not provide identifiers, we encourage authors to supply the search terms that will return the data. For data that have been obtained from publicly available sources, please provide a URL and the specific data product name in the data availability statement. Data with a DOI should be further cited in the methods reference section.

Point-by-point response to Reviewers

Dear Editors,
Dear Reviewers,

We sincerely value the time and effort that you have put into reading our submitted manuscript, and we are grateful for your highly insightful feedbacks and careful suggestions.

Here, we provide a point-by-point response to each of comments. We have divided all Reviewers' comments (reported **in bold**) into different sections in order to make the revision easier to follow. The modifications made in the revised version of the manuscript are reported in blue.

Reviewer #1 (Remarks to the Author):

I reviewed the research report manuscript titled: “The Vertical Position of Visual Information Conditions Spatial Memory Performance in Healthy Aging”. This study evaluates the impact of the vertical localization of items on source memory in aging, and essentially shows a significantly lower accuracy of source memory in older vs. younger adults for items encoded in the upper field, but not for items encoded in the lower field. This finding holds from two relevant and independent types of data analyses, namely a multinomial processing (MPT) modeling approach and a more classical linear approach based on estimating discrimination indices. To probe whether this main finding was merely emerging from visual impairment, the authors further tested whether the source memory accuracy on the upper-field-encoded items was related to visual acuity, contrast sensitivity, or kinetic perimetry. They found no evidence for that. Item memory as well as reaction times were also investigated, and presented an impact of age, but not of encoding visual field.

In general, this study was motivated by the idea of extending the exploration of the ‘vertical meridian asymmetry’ (VMA) phenomenon, by which numerous visual abilities differ between the upper and lower visual fields, to higher cognitive functions like memory, or more precisely here, to source memory. Importantly, this was also addressed regarding the impact of aging. In light of their findings, the authors argue that the age-related VMA observed here for spatial memory disfavoring the upper field is in fact independent from the previously shown VMA pertaining to vision, appears with age contrarily to visual VMA, and might interestingly originate from higher visual regions.

Overall, I found the manuscript very well written and justified, both theoretically and methodologically. The conclusion is quite original and has indeed the potential to be of interest for the community and the wider field. However, I believe that one important dimension of the experiment (‘the pre-experimental source expectation’, see major concern) still remains to be addressed for the conclusion to be convincing. On a less important note, I also think that the question of the ecological importance of memory for non-directly gazed items needs to be discussed more for the overall impact to be clear (see last minor comment).

We very much appreciate the reviewer's positive feedback and fully understand the concerns raised. We have strived to answer each comment as thoroughly as possible.

1 -- Major concern:

There is evidence that objects that are usually in further distances are usually seen in the upper field, and that those usually at close distances are usually seen in the lower field (see for instance Previc, 1990, cited in the manuscript). Based on that, it is reasonable to expect

that the judgment of the vertical location where an item was at encoding will be biased towards its usually seen vertical field. This 'pre-experimental source expectation' is key in the scope of the goals and claims of this study, but is however not explicitly controlled, nor discussed in the manuscript. I propose below a series of points to discuss, and analyses to run to address the consequent current uncertainty of the main claim.

While eliminating objects "strongly associated with the ground, the ceiling, or the sky" from the stimulus sets and counterbalancing items across participants as it was done here are already important careful controls, I would argue that these might not be sufficient in avoiding any bias relative to the overall pre-experimental source expectation.

Specifically, based on pre-experimental exposure in daily life, the objects chosen could for instance tend to be overall expected in the lower visual field. If this is the case, then the overall source memory responses might have been biased towards the response 'down', and thereby the accuracy towards being better on the lower-field-encoded items and worse on the upper-field-encoded-items.

Since it is expected that this bias would operate more in poorer memory situations and known that older adults are impaired on episodic memory such as source memory, a bias in the items pre-experimental source expectation could explain the main pattern of results, compromising the current claimed interpretation.

Ideally, it would have been interesting to separate the items in two controlled and balanced groups (one with pre-experimental source expectation in the upper visual field, and the other in the lower visual field) to evenly distribute the items in each condition and/or assess their putative impact.

Without this, it could still be interesting to assess this vertical visual field of the pre-experimental source expectation of each item used, and take this variable into account in the analyses (e.g., as a trial-level covariate in the analyses, and otherwise at the condition level). At least, it would be important here to report whether a bias in the source location reported exists on false alarms (responses "old" on 'new' items), i.e. whether the parameter 'g' (in the MPT modeling approach) and/or whether the number of source location of type 'down' reported on false alarms (in a more classical approach) are/is different than the chance level. Showing such bias on items never seen in the upper or lower visual field would disfavor the interpretation of an actual VMA for source memory (in line with its lack of relationship with participants visual VMA). On the contrary, the absence of such bias would suggest that the stimuli were well balanced in terms of pre-experimental source expectation, making the central claim of the manuscript stronger.

We thank the reviewer for this detailed and insightful comment. We agree that pre-experimental exposure to objects could influence the results pertaining to item and source memory. We had this caveat in mind when designing the experiment and this is why, as the reviewer points out, we made sure that the chosen items weren't too strongly associated with either the ground, the sky, or the ceiling. Across all participants, we made sure that each object was presented both in the upper portion and in the lower portion of the screen. That being said, we understand the reviewer's concern that these first-level controls may not be sufficient. We have therefore followed their advice and conducted additional analyses.

First, we assigned a pre-experimental source expectation to each item included in the experiment (i.e., upper expectation or lower expectation, see Figure 1 for two examples). Of note, all objects that are most frequently found at eye-level were assigned to the lower visual field.

Figure 1. Examples of items from the experiment with different source expectations. The umbrella is associated with the upper visual field while the wine glass is associated with the lower visual field.

We conducted a linear mixed model to examine the effects of position and age on the percentage of correct source answers adjusting for the pre-experimental source expectation of each object. We found no significant influence of the pre-exposure ($F(1, 2747) = 1.79, p = 0.18$), whereas the main effects of age and position of item on source memory remained significant (Figure 2).

Figure 2. Spatial memory performance according to the source expectation of objects in young and older participants. Each item was assigned the position in space (upper or lower) that it is most frequently associated with. Horizontal colored lines correspond to individual data points. The horizontal black lines correspond to the mean in each group. VF = visual field.

Second, as requested by the reviewer, we tested whether source decisions for false alarms were biased for the lower visual field. It should be underlined that the percentages of false alarms across the experiment were quite small in young adults ($M = 4.4; SD = 3.6$). This renders any statistical analysis on false alarms biased as it is driven by a handful of older participants. We computed the vertical meridian asymmetry (VMA) for false alarms (see below). A score of 0 indicates no difference between the proportion of upper and lower responses for false alarms. A score superior to 0 indicates a higher proportion of lower responses and a score inferior to 0 indicates a higher proportion of upper responses.

$$VMA = \frac{(lower\ false\ alarms - upper\ false\ alarms)}{mean(lower\ false\ alarms, upper\ false\ alarms)}$$

We conducted a linear regression to study the influence of age group on the VMA for false alarms (Figure 3). We found no significant difference in the VMA between young and older adults ($R^2 =$

0.021, $F(1, 42) = 1.92$, $p = 0.17$). Older and young adults had similar proportions of *upper* and *lower* responses for false alarms. Moreover, the VMA was not significantly different from 0 as indicated by one-sided sign tests in young ($M = 0.028$, $SD = 1.42$, $p = 0.41$) and older adults ($M = 0.55$, $SD = 1.01$, $p = 0.06$).

Overall, we conclude that there is no statistical evidence for a bias in the source location chosen by participants on false alarms. Note that we did not conduct further analyses on the g parameter estimated from the MPT model as it also estimates the probability of guessing “up” for items that were previously seen.

Figure 3. Vertical meridian asymmetry (VMA) for false alarms in young and older adults. A score of 0 indicates no difference between the proportion of upper and lower responses for false alarms. A score superior to 0 indicates a higher proportion of lower responses and a score inferior to 0 indicates a higher proportion of upper responses. Horizontal colored lines correspond to individual data points. Horizontal black lines correspond to the mean VMA in each group.

The absence of a source location bias for items that were never seen (“new” items for which participants responded “old”) implies that the stimuli were well balanced in terms of pre-experimental source expectation, and that it is unlikely to be the main explanation for our results. Furthermore, even if pre-exposure to items played a role, it wouldn’t explain why older adults specifically showed a spatial memory deficit in the upper visual field. Indeed, while participants from both age groups have the same visual field expectations of objects, they don’t exhibit the same pattern of impairment. One may wonder whether the greater experience with objects accumulated across the lifespan renders the spatial processing of objects in their “atypical” location (i.e., the upper visual field) more difficult. Interestingly, Kaiser and colleagues showed that the expected position of an object could enhance the processing of object identity but not of its location in space (Kaiser & al., 2018, *Neuroimage*). We believe that the reviewer’s comment opens up new avenues for future research.

Following another reviewer’s suggestion, we moved the standard Pr analysis from the Supplementary Information file to the main manuscript. Interestingly, results from this analysis may be interpreted in light of the current comment about pre-experimental source expectation. We found that both age groups exhibited a slight lower field benefit for item recognition. It could in fact be the case that because everyday objects are more frequently encountered in the lower visual field, encoding is facilitated for items presented in the lower portion of the screen. It has been proposed that the UVF advantage for sex-categorization of faces and hands may originate from the association between sex-relevant cues and the upper portion of stimuli (Quek & Finkbeiner, 2014, *Adv. Cogn. Psychol.*). Furthermore, a recent study revealed that visual experience could contribute to the formation of vertical biases (Tsurumi et al., 2022, *Dev. Sci.*). The authors

concluded that perceptual asymmetries reflect adaptations to typical locations in the surrounding environment.

In conclusion, while pre-experimental source expectation may contribute in part to the lower field advantage for item recognition, we demonstrated that the spatial memory deficit in older adults is likely attributable to an altogether different mechanism.

We added a paragraph in the Results section and two paragraphs in the Discussion bringing up this topic. We also included Figures 2 and 3 presented above as extended data. Figure 2 becomes Fig. E4, and Figure 3 becomes Fig. E5 in the revised manuscript.

Results section

- Lines 168 – 184: “Notably, we verified that the latter result could not be attributable to the expected position of objects in space. We first assigned a pre-experimental source expectation to each item included in the experiment (i.e., upper or lower expectation). We conducted a linear mixed model to examine the effects of position and age on the percentage of correct source answers adjusting for the pre-experimental source expectation of each object. We found no significant influence of the pre-exposure ($F(1, 2747) = 1.79, p = 0.18$), whereas the main effects of age and position of item on spatial memory remained significant (Fig. E4). Second, we also tested whether source decisions for false alarms were biased for the lower visual field. Indeed, a bias in source location response for items that were never seen could imply that the objects were not well balanced in terms of pre-experimental source expectation. We computed the VMA for false alarms and conducted a linear regression to study the influence of age group on the VMA for false alarms (Fig. E5). We found no significant difference in the VMA between young and older adults ($R^2 = 0.021, F(1,42) = 1.92, p = 0.17$). Older and young adults had similar proportions of upper and lower responses for false alarms. Moreover, the VMA was not significantly different from 0 as indicated by one-sided sign tests in young ($M = 0.028, SD = 1.42, p = 0.41$) and older participants ($M = 0.55, SD = 1.01, p = 0.06$). There is thus no statistical evidence for a bias in the choice of source location for false alarms.”

Discussion section

- Lines 267 – 276: “An effect of the objects’ vertical position on item memory was found across all participants. Object recognition and reaction times were improved for items presented in the lower portion of the screen. In the literature, object processing tasks including letter identification, target localization, word discrimination, and face-sex categorization, have been conferred an upper visual field advantage^{25,26,49,50}. One possibility is that visual experience with specific stimuli shapes the formation of vertical visual field biases. For example, researchers are evoking the possibility that faces are better encoded in the upper visual field because they are situated on the upper portion of the body^{31,51}. That everyday objects are more frequently encountered in the lower visual field may explain why the encoding of items in our task was facilitated in the lower portion of the screen for all participants.”

- Lines 278 – 284: “We revealed that older adults have significantly worse spatial memory than young adults but only for objects presented in the upper visual field. In other words, spatial memory may be preserved in aging for visual information situated in the lower field. Of note, while the expectation of the position of objects in space may contribute in part to the lower visual field advantage for item recognition, we demonstrated that the spatial memory deficit in older adults was most likely attributable to an altogether different mechanism.”

Minor comments:

2 -- Considering how the parameters ‘o’ and ‘g’ are defined in the MPT modeling approach (second paragraph of the Results section, and in Figure 2), my understanding is that they

don't correspond to the "probability of correctly guessing that an item was presented [in the upper visual field, for g]" as it is written along the manuscript, but instead to the "probability of guessing that an item was presented [in the upper visual field, for g]". Indeed, whatever the item (target or distractor), 'o' corresponds to the probability of guessing that the item is 'old' and not 'new', and 'g' to the probability of guessing that the item was presented in the upper ('up') and not lower ('down') visual field, such that, even if responding 'old' or 'up' would be incorrect on distractors, this would be more probable with 'o' or 'g' being higher. In the scope of being even more in line with how the parameters in the MPT approach here are defined (e.g., second paragraph of the Results section), since each parameter measures the "probability of occurrence" – rather than the "accuracy" – of a specific cognitive process, this also applies to the specific cognitive processes of 'guessing old' ('o') and 'guessing up' ('g'). This should be corrected along the manuscript (specifically in the Results and Methods sections).

We would like to sincerely thank the reviewer for pointing out these errors. Indeed, we wrongly defined parameters *o* and *g*. We have amended the definitions throughout the manuscript as follows.

Methods section

- Lines 490 – 492: "The parameter *o* corresponds to the probability of guessing that the object was presented (i.e., answering "old"), and the parameter *g* estimates the probability of guessing that the item was presented in the upper part of the screen, given that item recognition has failed."

Results section

- Lines 139 – 141: "We found that older adults' probability of guessing that an item was presented during the encoding phase was significantly lower than young adults'."
- Lines 204 – 205: "the guessing parameter *g*, defined as the probability of guessing that an item was presented in the upper visual field."
- Figure 2: "Parameter *o* refers to the probability of guessing that an item is "old" while parameter *g* refers to the probability of guessing that an item was presented in the upper visual field."

Extended data section

- Figure E1. "The parameter *o* describes the probability of guessing that the item was presented (i.e., answering "old"). The parameter *g* on the other hand describes the probability of guessing that the item was presented in the upper visual field."

3 -- It would be interesting, and more consistent with other graphs of the manuscript, to plot figures 2B and 2C in the same format than the others (e.g., figure 3), in particular to have the same visualization of the subject values distribution.

We understand that for visualization purposes it would be more suited to have the same type of plot throughout the manuscript. However, we think that using bean plots is somehow misleading for figures 2B and 2C. Indeed, it is more difficult to discern the differences in population distribution between S_{down} and S_{up} (Figure 4). Posterior densities allow the overlap between young and older distributions to be clearly visualized, and they are more adequate for plotting MPT parameter estimates. The package in R that we have used to fit MPT models, TreeBUGS, provides MPT-tailored functions to plot parameters adequately (Heck et al., 2018, *Behav. Res. Methods*). For the above reasons, we think it would be more appropriate to keep the original figure.

Figure 4. Amended figure (Fig. 2 in original manuscript) showing item and source memory performance in young and older participants. Bean plots were used instead of posterior density plots.

4 -- While the topic tackled in this study is interesting and promising for better understanding the cognitive impairment of aging and accompanying the aging population with age-friendly strategies like the authors underlined, it must be acknowledged that it only covers the case of source memory when the actual items are encoded in the absence of direct saccade on these. In real life though, it might be more ecologically the case that we actually look at the items – and thereby saccade to them – to encode them. Moreover, albeit of smaller importance, these results on non-directly-gazed items only apply to source memory and not item memory, meaning that they are more concerning situations of remembering the vertical location where an item – only seen in the periphery – was situated. I believe these aspects relative to the ecological importance of the results should be more explicitly discussed, and the impact of this result in daily life strategy mitigated and/or discussed accordingly.

We agree with the reviewer that these points should be highlighted in the discussion. First, the reviewer rightfully mentions that our results only cover the case in which objects are encoded in the absence of direct saccades towards them. One can't ignore the possibility that making a saccade towards an item could compensate, to a certain extent, for the poorer spatial memory performance in the upper visual field of older adults. However, recent studies shed light on the resilience of visual asymmetries even when freely exploring the surrounding space with eye movements (Hanning et al., 2022, *iScience*). Moreover, it has been suggested that regions of the brain with preferences for certain portions of the visual field (e.g., occipital place area) could actually be guiding gaze allocation (Malcolm et al., 2018, *Front. Hum. Neurosci.*). It may be the case that the spatial memory deficit in the upper visual field of older adults modifies older adults' sampling of natural scenes in free-viewing conditions. With the reviewer's comments in mind, we have mitigated the impact of our findings in the discussion.

Discussion section

- Lines 340 – 343: “We must stress that our results apply only to non-fixated objects and that an important next step in the field is to investigate whether free-viewing conditions can rescue older adults’ spatial memory deficit in the upper visual field. Finally [...]”

Second, we agree with the reviewer that our results only pertain to source memory and not to item memory. It could thus be important to better describe in the discussion how such a specific deficit could impact older adults’ daily life. We argue that spatial memory impairments in the upper visual field may be particularly detrimental for complex abilities such as spatial navigation or driving. Indeed, these skills require the processing of large landmarks or buildings (i.e., immovable objects), visual information that is most often found in the upper visual field (Groen et al. 2017, *Philos. Trans. R. Soc. Lond. B. Biol.*). We acknowledge that these aspects weren’t detailed enough in the discussion and we have thus sought to make them more explicit.

Discussion section

- Lines 334 – 336: “These abilities require the processing and spatial encoding of visual information that is most often found in the upper visual field: large immovable objects, road signs, monuments, or buildings.”

References

Groen, I. I. A., Silson, E. H. & Baker, C. I. Contributions of low- and high-level properties to neural processing of visual scenes in the human brain. *Philos. Trans. R. Soc. Lond. B. Biol. Sci.* (2017)

Hanning, N. M., Himmelberg, M. M. & Carrasco, M. Presaccadic attention enhances contrast sensitivity, but not at the upper vertical meridian. *iScience* (2022)

Heck, D. W., Arnold, N. R. & Arnold, D. TreeBUGS: An R package for hierarchical multinomial-processing-tree modeling. *Behav. Res. Methods* (2018)

Kaiser, D., Moeskops, M. M. & Cichy, R. M. Typical retinotopic locations impact the time course of object coding. *Neuroimage* (2018)

Malcom, G. L., Silson, E. H., Henry, J. H. & Baker, C. I. Transcranial magnetic stimulation to the occipital place area biases gaze during scene viewing. *Front. Hum. Neurosci.* (2018)

Reviewer #2 (Remarks to the Author):

The authors conducted an interesting study of the influence of target location in upper vs, lower visual field on subsequent item recognition memory and “source memory” for that item’s location. The underlying issue – whether more primary deficits in visual processing might be responsible or age-differences in episodic memory – is an important one, but I have some concerns about the study which I hope the authors might address.

1 -- Introduction: The authors provide an interesting list of tasks in which there are upper vs. lower visual field advantages. Perhaps the authors could comment on whether the distinction between tasks with upper and lower visual field advantages is related to whether attentional functions are more implicated (e.g., visual search, change detection) or less implicated (e.g., hue discrimination, contrast sensitivity). That might be relevant to the two mnemonic abilities the authors explore.

We agree with the reviewer that attentional functions are essential to consider when studying visual asymmetries.

As pointed out by the reviewer, some studies have tried to link the quantity of attentional resources required, and whether the processing advantage is in the upper or lower visual field. Indeed, it has long been thought that attentional demands may explain perceptual asymmetries (Danckert & Goodale, 2003, In *S.H. Johnson-Frey*). For example, Kraft and colleagues (2011, *Hum. Brain. Mapp.*) concluded that a lower visual field advantage is frequently apparent in tasks requiring stationary attention (e.g., reaching, grasping) while an upper visual field advantage is found when the attentional focus is frequently shifted (e.g., visual search).

However, conflicting evidence has since been provided by many researchers. The current consensus seems to be that visual asymmetries along the horizontal and vertical dimensions are actually resilient to endogenous, exogenous, and pre-saccadic attention (Hanning et al., 2022, *iScience*; Purokayastha et al., 2021, *Atten. Percept. Psychophys.*; Roberts et al., 2016, *J. Vis.*). Levine and McAnany (2005, *Vis. Res.*) argued that if the difference between upper and lower visual field advantages depended upon attention, more cognitively demanding tasks should be better performed in the lower visual field, where the ability to finely focus attention is best. They nonetheless show that it is not the case.

Taking the latter into account, we think that delving into too much detail about attentional functions and vertical asymmetries may be out of the scope of this manuscript. We nonetheless appreciate the reviewer’s remark and we have added a few references related to attention in the revised Introduction.

Introduction section

- Lines 31 – 33: “These asymmetries exist across a vast array of stimuli orientations, sizes, and luminance levels, and they appear resilient to endogenous, exogenous, and presaccadic attention¹⁰⁻¹².”

2 -- Methods: Did the eye tracking data indicate how well participants were actually able to maintain fixation on the central cross during the lengthy (2500 ms) presentation of the stimuli? Was there the same degree of “improper” saccades to the object pictures for upper and lower offsets?

We did perform an analysis to check how well participants were able to maintain fixation on the central cross during the encoding phase. We used the 2/4 method and corresponding Fujii

classification as a criterion for inclusion (Fuji et al., 2002, *Ophthal.*). Participants who were found to have less than 70% of their fixations inside of a 4° circle around the central cross were deemed to have unstable fixation and were thus excluded from the analysis.

Participants who were not excluded had relatively stable fixation. On average across the experiment, young participants had 95.8% of their fixations within a 4° circle of the central fixation point. Older adults also had a surprisingly stable gaze with 80.7 % of their fixations within this 4° circle. The eye-tracking statistics for included participants are in stark contrast with the 6 excluded older adults who couldn't maintain a stable gaze. They had, on average, 38.2% of their fixations that fell within the 4° circle.

Notably, we noticed an error in our definition of the Fuji classification in the Methods section, we modified the manuscript as follows.

Methods section

- Lines 439 – 440: “The latter method considers participants to have unstable fixation if less than 75% of their fixation points are **inside** a 4° circle.”

The reviewer subsequently asked about quantifying “improper” saccades during the encoding phase which we found to be highly relevant. For all participants, we extracted the saccades that were performed outside of the 4° circle around the central cross. We categorized them as being directed towards the upper portion or the lower portion of the screen. Then we computed the vertical meridian asymmetry (VMA) for improper saccades using the formula below. A score of 0 indicates no difference between the proportion of upper- and lower-directed saccades. A score superior to 0 indicates a higher proportion of lower-directed saccades and a score inferior to 0 indicates a higher proportion of upper-directed saccades.

$$VMA = \frac{(lower\ field\ directed\ saccades - upper\ field\ directed\ saccades)}{mean(lower\ field\ directed\ saccades, upper\ field\ directed\ saccades)}$$

First, we found that both young and older adults’ mean VMA was around 0 (young adults: M = -0.15, SD = 0.98; older adults: M = -0.028, SD = 0.57), signifying that they made an equivalent number of improper saccades to the upper and lower parts of the screen. We then conducted a linear regression to test the influence of age group on this VMA (Figure 1). We found no significant influence of age group on the VMA ($R^2 = -0.018$, $F(1, 42) = 0.25$, $p = 0.62$).

Figure 1. Vertical meridian asymmetry (VMA) for “improper saccades” during the encoding phase in young and older adults. Horizontal colored lines correspond to individual data points. Horizontal black lines correspond to the mean VMA in each group.

Considering the fact that there doesn't seem to be any apparent biases in the direction of improper saccades during the encoding phase, we have decided not to include this analysis in the manuscript.

3 -- Methods: Why were participants only tested on 20 studied items (along with 10 foils) out of the 30 that they had studied in each run?

This is a relevant point that we acknowledge should have been better explained in the manuscript. The design of our paradigm was adapted from the mnemonic similarity task (Stark et al., 2019), a widely used tool for assessing memory functions in healthy and pathological populations. In this task, the same number of images are presented in the encoding and test phases. The latter consists of 1/3 new images, 1/3 old images and 1/3 "lure" images. In a similar vein, we chose to include 1/3 old images that were presented in the upper visual field, 1/3 old images that were presented in the lower visual field, and 1/3 new images.

Moreover, we had the constraint that the task couldn't be too lengthy for older participants. Including all 30 old images in the test phase would have considerably increased task duration. Studies on recognition memory in healthy aging frequently choose to include only a portion of the encoded objects in the test phase (Grady et al., 2005, *Neuropsychologia*; Howard et al., 2006, *Psychol. Aging*; Springer et al., 2005, *Neuropsychology*).

We have added a sentence in the Methods sections justifying our choice.

Methods section

- Lines 397 – 399: "Only two-thirds of the encoded objects were presented during the test phase in order to limit task duration for older adults."

4 -- Results: While the MPT analysis employed has its advantages, presenting the results only in terms of the parameters of the model is opaque to any readers who are not familiar with the method. The authors should more fully explain the exact meaning of what they are presenting in the figures. Additionally, before presenting the results in terms of the MPT models, the authors should present the raw accuracy data.

We understand the reviewer's point of view and we agree that the results from the MPT analysis can be somewhat opaque. In accordance, we added the raw accuracy data as suggested. We also reasoned that the manuscript would be improved by moving the standard *Pr* analysis, previously presented in the Supplementary Information file, to the main text. Confronting the two types of analyses directly may equip the reader with the necessary tools to better understand the Results section. We therefore modified the Methods and Results sections.

The raw accuracy data (hits, false alarms, correct rejections) in young and older adults was added as a table in the Extended Data.

Results section

- Lines 85 – 86: "Raw accuracy data for young and older participants are presented in the extended data (Table E1)."

Extended data section

- Table E1:

Group	Hits	False Alarms	Correct Rejections
	Rate (M ± SD)	Rate (M ± SD)	Rate (M ± SD)
Young	81.5 % ± 14.6 %	4.4 % ± 3.6 %	93.3 % ± 5.1 %
Older	69.1 % ± 17.3 %	10.2 % ± 9.0 %	82.1 % ± 17.4 %

- Table E1 legend: “Summary of young and older participants’ accuracy scores on the object recognition paradigm. Hit rate, false alarm rate, and correct rejection rate are presented. M: mean; SD: standard deviation.”

The paragraph explaining the standard *Pr* analysis, initially included in the Supplementary Information file, was added to the Methods section of the main text.

Methods section

- Line 442 (Subtitle): “Standard *Pr* and reaction times analysis”
- Lines 443 – 462: “To assess participants’ item and source memory, the literature uses a discrimination index or “*Pr*” which is a measure of accuracy developed by Snodgrass & Corwin (1988)⁷³ for such paradigms. For item memory, the *Pr* was calculated by subtracting the proportion of false alarms to the proportion of hits. For spatial memory, the *Pr* was calculated by subtracting the number of incorrect spatial judgments to the number of correct spatial judgments and dividing by 160 (= the total number of source trials that there should be if a participant scored perfectly on the item memory task) in order to account for the interindividual variability in the number of source trials. If a participant frequently answered that an item was “New”, they would have far less source trials than a participant who frequently answered that an item is “Old”. Reaction times were recorded for both item and spatial memory. They corresponded to the elapsed time with the presentation of the object and the button press. Two linear mixed models were used to evaluate the influence of age group and initial object position on item *Pr* and source *Pr*. The random-effects structure was chosen to best fit the experimental design while remaining parsimonious⁷⁴. The two models thus contained subjects as random intercepts. The reliability of each fixed effect was tested using the Akaike information criterion (AIC) goodness-of-fit statistic, comparing one model which included the effect of interest and another which didn’t. Using this procedure, age group and object position were kept as fixed effects after determining that the inclusion of sex and block sequence did not improve model fits. The normality of residuals of the two linear mixed models were carefully inspected using residual plots.”

We then separated the results pertaining to item and spatial memory into two subparagraphs: one for the standard analysis and one for the MPT analysis. Figures were amended accordingly. We divided Figure 2 from the original manuscript into 3 new individual figures. The new Figure 2 now illustrates only the MPT binary trees that were used to model data and is accompanied by a legend that strives to better explain MPT modeling.

- New Figure 2:

- New Figure 2 legend: “Graphical representation of the MPT model used to analyze the experimental data. A tree is used to model experimental data. Colored rounded rectangles represent the three different trial types: the object was presented in the upper part of the screen (“UP”), in the lower part of the screen (“DOWN”), or was not presented (“NEW”). Grey rounded rectangles represent participants’ possible answers. I_{up} and I_{down} are the probabilities of remembering an item that was presented in the upper visual field and lower visual field, respectively. S_{up} and S_{down} are the probabilities of remembering the position of an item that was presented in the upper visual field and lower visual field, respectively. Parameter o refers to the probability of guessing that an item is “old”, while parameter g refers to the probability of guessing that an item was presented in the upper visual field.”

The new Figure 3 shows results from the standard and MPT analyses related to item memory. The legend was lengthened in order to include a clearer description of how the MPT results can be interpreted. The main text was also modified.

- New Figure 3:

- New Figure 3 legend: “Behavioral performance on the item memory task as evaluated by **(A)** standard *Pr* analyses and **(B)** MPT analyses. **(A)** Item memory is evaluated during the object recognition task in which subjects answer the question “*Is the object old or new?*”. Item *Pr* corresponds to the percentage of good answers minus the percentage of false alarms. Horizontal colored lines correspond to individual data points. Horizontal black lines correspond to the mean in each group. * $p < 0.05$ **(B)** Group-level posterior distributions for parameters related to item memory. The posterior distribution corresponds to updated knowledge about the parameters l_{up} and l_{down} after considering the current data. l_{up} is the probability of remembering an item that was presented in the upper part of the screen and l_{down} is the probability of remembering an item that was presented in the lower part of the screen.”
- Line 104 (Subtitle): “Relationship between item memory and objects’ vertical position”
- Lines 105 and 131 (Subtitles): “Standard analysis” and “MPT analysis”
- Lines 106 – 130: “We conducted a linear mixed model to investigate the effects of age group and object position on item *Pr*. The fit of the model was not improved by adding the interaction between age group and object position (with interaction: AIC = -150.91, without interaction: AIC = -152.69). The interaction was therefore not retained in subsequent analyses. We found a significant main effect of age group on item *Pr* ($F(1, 43) = 17.66, p = 0.00013$). On average, item *Pr* was higher in young adults ($M = 0.77, SD = 0.14$) than in older adults ($M = 0.59, SD = 0.16$) suggesting that young adults had better object recognition memory than older adults (Fig. 3A). We also observed a significant main effect of object position on item *Pr* ($F(1, 44) = 11.58, p = 0.0014$). Overall, participants were slightly better at recognizing objects that were presented in the lower visual field

($M = 0.71$, $SD = 0.18$) than in the upper visual field ($M = 0.67$, $SD = 0.17$). We conducted an additional linear mixed model that examined the influence of age group and object position on the distribution of misses. We found a main effect of age on the number of misses ($F(1, 43) = 4.40$, $p = 0.042$), revealing that older adults missed more item memory trials ($M = 4.57$, $SD = 6.88$) than did young adults ($M = 1.87$, $SD = 3.27$). No effect of object position was uncovered ($F(2, 88) = 1.40$, $p = 0.25$), indicating that missed trials were equivalently distributed across up, down, and new objects.

Using a gamma generalized linear mixed model, we found a significant main effect of age ($\chi^2(1) = 24.31$, $p < 0.0001$) on reaction times during the item memory task (Fig. 5). This result reveals that older adults were slower ($M = 1.28s$, $SD = 0.46s$) to respond than young adults ($M = 1.01s$, $SD = 0.44s$). We also observed a main effect of object position ($\chi^2(1) = 5.69$, $p = 0.017$): all participants were slightly slower in reacting to objects that had appeared in their upper visual field ($M = 1.14s$, $SD = 0.47s$) than in their lower visual field ($M = 1.12s$, $SD = 0.47s$). Finally, we showed that the block number had an impact on reaction times ($\chi^2(7) = 343.77$, $p < 0.0001$), suggesting that both young and older participants improved their performance throughout the experiment (Fig. E3)."

The new figure 4 shows results from the standard and MPT analyses related to spatial memory. The legend was lengthened in order to include a clearer description of how the MPT results can be interpreted. The main text was also modified. Taking into account another reviewer’s remark, we added a control analysis to verify whether the spatial memory deficit in older adults could be attributed to the expectation that objects are situated in the lower field.

- New Figure 4:

- New Figure 4 legend: “Behavioral performance on the spatial memory task as evaluated by (A) standard *Pr* analyses and (B) MPT analyses. (A) Spatial memory is evaluated during the spatial

memory task in which subjects answer the question “*Was the object presented in the upper or lower visual field?*”. Source Pr corresponds to the percentage of correct answers weighted by the number of answered trials. Horizontal colored lines correspond to individual data points. Horizontal black lines correspond to the mean in each group. $*p < 0.05$. **(B)** Group-level posterior distributions for parameters related to spatial memory. The posterior distribution corresponds to updated knowledge about the parameters S_{up} and S_{down} after considering the current data. S_{up} is the probability of remembering the position of an item that was presented in the upper part of the screen, and S_{down} is the probability of remembering the position of an item that was presented in the lower part of the screen.”

- Lines 155 and 197 (Subtitles): “Standard analysis” and “MPT analysis”
- Lines 156 – 196: “We conducted a second linear mixed model analysis to investigate the effects of age group and item position on source Pr . Here, the fit of the model was improved by adding the interaction between age group and object position (with interaction: AIC = -39.08, without interaction: AIC = -35.79). The interaction was retained in subsequent analyses. We found a significant main effect of age group on source Pr ($F(1, 43) = 16.78, p = 0.00018$). On average, source Pr was higher in young adults ($M = 0.60, SD = 0.04$) than in older adults ($M = 0.33, SD = 0.04$), highlighting better spatial memory performance in young adults compared with older adults (Fig. 4A). While we did not observe a significant main effect of object position on source Pr , we found the interaction between age group and object position to be significant ($F(1, 43) = 5.37, p = 0.025$). Older participants showed poorer performance when prompted on the position of items that were presented in the upper visual field ($M = 0.33, SD = 0.18$) compared with objects that were presented in the lower visual field ($M = 0.47, SD = 0.20$).

Notably, we verified that the latter result could not be attributable to the expected position of objects in space. We first assigned a pre-experimental source expectation to each item included in the experiment (i.e., upper or lower expectation). We conducted a linear mixed model to examine the effects of position and age on the percentage of correct source answers adjusting for the pre-experimental source expectation of each object. We found no significant influence of the pre-exposure ($F(1, 2747) = 1.79, p = 0.18$), whereas the main effects of age and position of item on spatial memory remained significant (Fig. E4). Second, we also tested whether source decisions for false alarms were biased for the lower visual field. Indeed, a bias in source location response for items that were never seen could imply that the objects were not well balanced in terms of pre-experimental source expectation. We computed the VMA for false alarms and conducted a linear regression to study the influence of age group on the VMA for false alarms (Fig. E5). We found no significant difference in the VMA between young and older adults ($R^2 = 0.021, F(1,42) = 1.92, p = 0.17$). Older and young adults had similar proportions of upper and lower responses for false alarms. Moreover, the VMA was not significantly different from 0 as indicated by one-sided sign tests in young ($M = 0.028, SD = 1.42, p = 0.41$) and older participants ($M = 0.55, SD = 1.01, p = 0.06$). There is thus no statistical evidence for a bias in the choice of source location for false alarms.

Studying reaction times during the spatial memory task, we found a significant main effect of age on reaction times ($\chi^2(1) = 21.83, p < 0.0001$; Fig. 5), meaning that older adults were slower ($M = 1.26s, SD = 1.16s$) to respond than young adults ($M = 0.69s, SD = 0.81s$). Despite the performance difference reported above, object position did not affect reaction times during the spatial memory task ($\chi^2(1) = 0.13, p = 0.72$). This finding highlights that participants responded equally rapidly to objects presented in the upper or in the lower visual field. We also showed a significant main effect of block number ($\chi^2(7) = 867.37, p < 0.0001$) as well as a significant effect of the interaction between age group and block number on reaction times ($\chi^2(7) = 39.18, p < 0.0001$). Although all participants improved their reaction times on the spatial memory task across blocks, older adults

showed a steeper learning curve (*first block*: $M = 2.06s$, $SD = 1.58s$; *last block*: $M = 0.95s$, $SD = 0.88s$) than young adults (*first block*: $M = 0.99s$, $SD = 0.93s$; *last block*: $M = 0.60s$, $SD = 0.80s$; Fig. E3)."

The Introduction was modified in accordance with the inclusion of this new analysis.

Introduction section

- Lines 63 – 68: "We computed the probabilities of remembering objects and their spatial location as a function of upper or lower visual field presentation positions using *standard Pr analysis* and multinomial processing tree (MPT) modeling. *The two types of analyses are complementary as the first is commonly used in the literature and easily interpretable and the second, although more complex, has the advantage of making explicit the assumptions about the relationship between item and spatial memory.*"

Finally, we took out the last paragraph of the Results section from the original manuscript. It mentioned that we conducted standard *Pr* analyses, and that we included the results in the Supplementary Information File ("We also conducted standard linear *Pr* analyses in order to compare the results with those obtained from MPT modeling [...]"). Accordingly, the supplementary file was entirely removed from the new version of the manuscript.

We thank the reviewer for this insightful comment which has made us rethink the structure of our manuscript.

5 -- Results: While the statistics indicate non-significant differences between the parameter estimates of younger and older adults for I-down and S-down conditions, with some overlap in the distributions, it does not seem reasonable to categorically state that the groups do not differ, relative to the clearer overlap apparent in I-up. I would like to see the results of standard simple analyses of d' for items and location remembering for correct endorsed items. Despite its disadvantages, those simpler analyses do not make inherent assumptions regarding dual-process or single-process bases of the data, and it seems to me important to present those analyses along with the MPT parameter analyses.

We understand the reviewer's point of view and we have thus strived to improve the clarity of our analyses. As mentioned above, the Standard *Pr* analysis that was initially presented in the Supplementary Information file was moved to the main manuscript (see **Comment 4**).

The standard analysis sheds light on two additional aspects to the data. First, it appears that item recognition, as measured with Item *Pr*, is impaired in older adults compared with young adults. This result stands in stark contrast with the MPT analysis based on the Source-Item model of memory that reveals no age-related differences in item memory. The finding that traditional recognition memory is impaired in older adults is in line with recent meta-analyses showing that age-related differences in item recognition are undeniably larger than zero (Fraundorf et al., 2019, *Psychol. Bull.*; Rhodes et al., 2019, *Psychon. Bull. Rev.*). We believe that the opposite pattern of results from the standard and MPT analyses can be interpreted in light of the specific definitions given to I_{down} and I_{up} by the model. In the Source-Item model, item memory refers to the probability of remembering an item given that the spatial position has not been recalled. Item memory thus seems more akin to a global measure of memory strength (i.e., familiarity). A different memory process could be measured by the Item *Pr* as many researchers have shown that item and source memory are very complicated to disentangle using traditional measures of accuracy (Batchelder & Riefer, 1990, *Psychol. Rev.*). Item *Pr* could thus reflect the detailed retrieval of past information (i.e., recollection). It is interesting to note that familiarity and recollection are differentially impacted by aging (Koen & Yonelinas, 2016, *Memory*). Familiarity processes remain intact in older age.

Second, the standard analysis revealed that, overall, participants are better at recognizing objects that appeared in their lower visual field. This finding is in line with our analyses showing that reaction times were slightly improved for items presented in the lower visual field. In the literature, object processing is usually found to be better performed in the upper visual field. Indeed, letter identification, target localization, word discrimination, and hand and face sex-categorization are tasks that have been found to have an upper visual field advantage (Hagenbeek & Van Strien, 2002, *Brain Cogn.*; Feng & Spence, 2014, *Perception*; Goldstein & Babkoff, 2001, *Neuroimage*; Quek & Finkbeiner, 2014a, *Cogn. Affect. Behav. Neurosci.*; Quek & Finkbeiner, 2014b, *Adv. Cogn. Psychol.*). Notably, these behavioral tasks require perceptual and attentional capacities rather than mnemonic abilities. Moreover, they used non-naturalistic stimuli in the form of gratings or blocks or human body parts such as faces and hands. No previous study has looked at vertical asymmetries in the recognition of everyday objects. The fact that everyday objects are more frequently encountered in the lower visual field may facilitate the encoding of items presented in the lower portion of the screen. It has indeed been proposed that the UVF advantage for sex-categorization of faces and hands may originate from the association between sex-relevant cues and the upper portion of stimuli (Quek & Finkbeiner, 2014b, *Adv. Cogn. Psychol.*). Furthermore, a recent study revealed that visual experience could contribute to the formation of vertical biases (Tsurumi et al., 2022, *Dev. Sci.*). The authors concluded that perceptual asymmetries reflect adaptations to typical locations in the surrounding environment.

We modified the abstract and discussion in accordance with the two findings uncovered by the standard *Pr* analysis.

Abstract

Lines 6 – 9: “Using standard *Pr* analysis and multinomial processing tree modeling, we revealed that although familiarity-based item memory remained intact in older age, spatial memory was impaired for objects presented in the upper visual field. Spatial memory in aging is therefore conditioned by the vertical position of information.”

Discussion section

- Lines 244 – 246: “The findings revealed a general lower field advantage for item recognition and an age-related decline in spatial memory for objects presented in the upper visual field only.
- Lines 249 – 250: “While we observed that spatial memory was consistently impaired in older adults, the results pertaining to item memory did not converge across analysis methods.”
- Lines 255 – 257: “A long line of research asserts that episodic recollection is impaired whereas familiarity and gist processing are aspects of item memory that remain intact in older age.”
- Lines 260 – 266: “The Item *Pr* measure of accuracy from the standard analyses, on the other hand, could reflect a recollection mechanism that involves a more detailed retrieval of past information. Indeed, previous research has highlighted the impossibility for item and spatial memory to be precisely disentangled using traditional statistical methods (Batchelder & Riefer, 1990). Our study, therefore, confirms that the general memory of an object is preserved and that its detailed recollection is impacted in healthy aging.”
- Lines 267 – 278: “An effect of the objects’ vertical position on item memory was found across all participants. Object recognition and reaction times were improved for items presented in the lower portion of the screen. In the literature, object processing tasks including letter identification, target localization, word discrimination, and face sex-categorization have been conferred an upper visual field advantage^{25,26,49,50}. One possibility is that visual experience with specific stimuli shapes the formation of vertical visual field biases. For example, researchers are evoking the possibility that

faces are better encoded in the upper visual field because they are situated on the upper portion of a body^{31,51}. That everyday objects are more frequently encountered in the lower visual field may explain why the encoding of items in our task was facilitated in the lower portion of the screen for all participants. Regarding spatial memory, our study provides nuance to the widespread view that older adults are impaired at recalling spatial contextual information.”

- Lines 288 – 291: “Importantly, the position of information **interacted with age for spatial memory performance** but not **for item recognition**. Such a result strongly implies that the observed upper visual field deficit in spatial memory does not arise from a biased distribution of covert attentional processes, **or the same deficit would have emerged in young adults.**”
- Lines 299 – 300: “The underpinning biological **bases** for the reported **lower visual field benefit in item recognition** and the upper visual field spatial memory deficit in older adults **are** unknown.”
- Lines 312 – 314: “It is therefore likely that top-down influences from higher visual regions play a role in **the general lower field preference for item recognition** and in the age-related spatial memory decline specific to the upper visual field.”
- Lines 319 – 322: “The slight lower field benefit for item memory in young and older adults is in line with fMRI studies reporting that the lateral occipital visual area (LO), an object-selective region, has stronger BOLD responses to objects in the lower than in the upper visual field⁵⁷⁻⁵⁹.”

6 -- Results: Another aspect of the data that is not transparent in the reported parameters is whether those parameters reflect the discrimination accuracy in each visual field, or the degree of endorsement of stimuli presented in that VF? In other words, does it reflect something like hit rate, or something like d'? I guess I'm wondering whether older adults had poorer performance in the UVF because of bias towards responding "lower". I'm a little confused here, because spatial memory ability is confounded with the location of the stimulus presentation. Had the authors presented objects in 4 quadrants, and then asked whether the target appeared on the right or the left, and found that discrimination was different for older and younger adults only in the upper VF, that would have been a clearer indication of different abilities for upper and lower VF presentations. But since in the actual paradigm any miss from UVF is a false alarm for LVF, how can that be unentangled to say that memory was better in LVF?

We understand the reviewer's concern regarding what the parameters mean exactly. First, we wish to underline that the spatial memory decline in the upper visual field of older adults was uncovered by both standard and MPT analyses. These concordant results strongly strengthen the case for an upper visual field spatial memory deficit in healthy aging.

The standard analysis uses a measure of source memory called the *Pr*. It is defined as the number of correct source judgments minus the number of incorrect judgments divided by the number of item hits. This measure has been criticized because its estimation is influenced by recognition performance and guessing biases (Batchelder & Riefer, 1990, *Psychol. Rev.*; Cooper et al., 2017, *Cortex*; Erdfelder et al., 2009, *J. Psychol.*). Including only the standard analysis in the manuscript would have indeed required further investigation in order to assert that the upper field decline in older adults didn't stem from a bias towards responding "lower". The caveats associated with the traditional *Pr* analysis prompted us to conduct the MPT modeling.

According to the Source-Item MPT model, S_{up} and S_{down} are defined as the probabilities of remembering the position of items that were presented in the upper or lower visual field, respectively. These parameters reflect the "probability of occurrence" of specific cognitive

mechanisms rather than accuracy measures. In fact, the interest of such models lies in the idea that one can disentangle individual cognitive processes. A potential bias towards responding “lower” or “upper” is modeled by parameter g . The latter corresponds to the probability of guessing that an item appeared in the upper portion of the screen. We found no significant difference in this parameter between young and older adults, suggesting that a bias for responding “lower” cannot explain the upper field decline in spatial memory in older age.

Finally, we would like to respond to the reviewer regarding their left/right discrimination paradigm idea. Although very interesting, it would have been difficult to implement such a task to answer the current scientific question. Perceptual asymmetries are also present along the horizontal dimension, which would have confounded our results. Moreover, it is well documented that vertical asymmetries are highest along the vertical meridian (Abrams et al., 2012, *Vision Res.*). For a first study looking into vertical visual asymmetries in aging, we reasoned that presenting objects along the meridian was the most prudent option.

7 -- Results: I also find some of the reported results non-intuitive, for example, the parameter estimate indicating that “older adults’ probability of correctly guessing that an item was presented during the encoding phase was significantly lower than young adults” – does that mean that they guessed to the same degree, but were less successful? Or that they simply guessed less? Since this is a contention based on a parameter extracted from the model, it might not correspond to actual cognitive processes. Perhaps the authors can explain how this emerged from their models.

We would like to thank the reviewer for pointing out these non-intuitive results. We realized that we wrongly defined parameters o and g in the original manuscript. Parameter o actually corresponds to the probability of guessing that the object was presented (i.e., participant answered “old”). Parameter g , on the other hand, estimates the probability of guessing that an item was presented in the upper visual field. We found a significant age-related difference in parameter o but not in parameter g . Older adults’ probability of guessing that an item is “old” was significantly lower than young adults’. In other words, in the event that neither the item nor its position in space is remembered, young adults will answer that the item is “old” to a greater extent than will older adults.

We amended the definitions for parameters g and o throughout the manuscript, and we added a more comprehensive explanation of the age-related difference for parameter o .

Methods section

- Lines 490 – 492: “The parameter o corresponds to the probability of guessing that the object was presented (i.e., answering “old”), and the parameter g estimates the probability of guessing that the item was presented in the upper part of the screen, given that item recognition has failed.”

Results section

- Lines 139 – 143: “We found that older adults’ probability of guessing that an item was presented during the encoding phase was significantly lower than young adults’ [...]. In other words, in the event that neither the object nor its position in space is remembered, young adults will answer that the item is “old” to a greater extent than will older adults.”
- Lines 204 – 205: “the guessing parameter g , defined as the probability of guessing that an item was presented in the upper visual field, was equivalent in young and older subjects.”
- Figure 2: “Parameter \$o\$ refers to the probability of guessing that an item is “old” while parameter \$g\$ refers to the probability of guessing that an item was presented in the upper visual field.”

Extended data section

- Figure E1. “The parameter o describes the probability of guessing that the item was presented (i.e., answering “old”). The parameter g on the other hand describes the probability of guessing that the item was presented in the upper visual field.”

References

- Abrams, J., Nizam, A. & Carrasco, M. Isoeccentric locations are not equivalent: the extent of the vertical meridian asymmetry. *Vision Res.* (2012)
- Batchelder, W. H. & Riefer, D. M. Multinomial processing models of source monitoring. *Psychol. Rev.* (1990)
- Cooper, E., Greve, A. & Henson, R. N. Assumptions behind scoring source versus item memory: Effects of age, hippocampal lesions and mild memory problems. *Cortex* (2017)
- Danckert, J. A. & Goodale, M. A. Ups and downs in the visual control of action. In S. H. Johnson-Frey (Ed.), *Taking action: Cognitive neuroscience perspectives on intentional acts.* (2003)
- Erdfelder, E., Auer, T-S., Hilbig, B. E., Abfal, A., Moshagen, M. & Nadarevic, L. Multinomial processing tree models: A review of the literature. *J. Psychol.* (2009)
- Feng, J. & Spence, I. Upper visual field advantage in localizing a target among distractors. *I-Perception* (2014)
- Fraundorf, S. H., Houridan, K. L., Peters, R. A. & Benjamin, A. S. Aging and recognition memory: A meta-analysis. *Psychol. Bull.* (2019)
- Fujii, G. Y., de Juan Jr, E., Sunness, J., Humayun, M. S., Pieramici, D. J. & Hang, T. S. Patient selection for macular translocation surgery using the scanning laser ophthalmoscope. *Ophtal.* (2002)
- Goldstein, A. & Babkoff, H. A comparison of upper vs. lower and right vs. left visual fields using lexical decision. *Neuroimage* (2001)
- Grady, C. L., McIntosh, A. R. & Craik, F. I. M. Task-related activity in prefrontal cortex and its relation to recognition memory performance in young and old adults. *Neuropsychologia* (2005)
- Hagenbeek, R. F. & Van Strien, J. W. Left-right and upper-lower visual field asymmetries for face matching, letter naming, and lexical decision. *Brain Cogn.* (2002)
- Hanning, N. M., Himmelberg, M. M. & Carrasco, M. Presaccadic attention enhances contrast sensitivity, but not at the upper vertical meridian. *iScience* (2022)
- Howard, M. W., Bessette-Symons, B., Zhang, Y. & Hoyer, W. J. Aging selectively impairs recollection in recognition memory for pictures: Evidence from modelling and ROC curves. *Psychol. Aging* (2006)
- Koen, J. D. & Yonelinas, A. P. Recollection, not familiarity, decreases in healthy ageing: Converging evidence from four estimation methods. *Memory* (2016)

Kraft, A., Sommer, W. H., Schmidt, S. & Brandt, S. A. Dynamic upper and lower visual field preferences within the human dorsal frontoparietal attention network. *Hum. Brain. Mapp.* (2011)

Levine, M.W. & McAnany, J. J. The relative capabilities of the upper and lower visual hemifields. *Vis. Res.* (2005)

Purokayastha, S., Roberts, M. & Carrasco, M. Voluntary attention improves performance similarly around the visual field. *Atten. Percept. Psychophys.* (2021)

Quek, G. L. & Finkbeiner, M. Face-sex categorization is better above fixation than below: evidence from reach-to-touch paradigm. *Cogn. Affect. Behav. Neurosci.* (2014a)

Quek, G. L. & Finkbeiner, M. Gaining the upper hand: evidence of vertical asymmetry in sex-categorisation of human hands. *Adv. Cogn. Psychol.* (2014b)

Rhodes, S., Greene, N. R. & Naveh-Benjamin, M. Age-related differences in recall and recognition: a meta-analysis. *Psychon. Bull. Rev.* (2019)

Roberts, M., Cymerman, R., Smith, R. T., Kiorpes, L. & Carrasco, M. Covert spatial attention is functionally intact in amblyopic human adults. *J. Vis.* (2016)

Springer, M. V., McIntosh, A. R., Winocur, G. & Grady, C. M. The relation between brain activity during memory tasks and years of education in young and older adults. *Neuropsychology* (2005)

Stark, S. M., Kirwan, C. B. & Stark, C. E. L. Mnemonic similarity task: a tool for assessing hippocampal integrity. *Trends Cogn. Sci.* (2019)

Tsurumi, S., Kanazawa, S., Yamaguchi, M. K. & Kawahara, J. Development of upper visual field bias for faces in infants. *Dev. Sci.* (2022)

Reviewer #3 (Remarks to the Author):

The authors use a MPT approach to compare spatial and item memory performance in younger and older adults as a function of whether stimuli were presented in the upper or lower hemifield during study. Using a Source-Item model introduced by Cooper et al. (2017), the authors conclude that item memory is invariant across upper and lower positions for both young and old participants, but that spatial memory is impaired in older participants for list items presented in the upper visual field.

This study tackles an interesting question using a principled theoretical analysis. The specific Source-Item model used here has been criticized by Kellen and Singmann (2017) on the grounds of parameter polysemy, which is troubling. This means that common parameters appearing in different processing trees (e.g., Up/Down vs. New, from Figure 2) are defined slightly differently and reflect slightly different psychological processes. Further, the analyses are conducted in a way that leaves a lot of information unexamined. This information, I believe, leads to an interpretation of the data that is potentially quite different from what the authors conclude based on the current analyses. I am therefore concerned that the conclusions do not necessarily follow from the data. Minimally, there are some potential caveats and qualifications that might need to be made. I detail my concerns below.

Major concerns:

1 -- Given the importance of good vision in the task, I was surprised that participants requiring glasses were not screened out. What proportion of younger/older participants who normally wear glasses completed the task without them? If these participants are unevenly distributed, or the visual deficit corrected by glasses is larger in one group vs. another, might this explain an apparent spatial memory deficit (e.g., due to greater variability in encoding location information)?

We thank the reviewer for their comment. Good vision is indeed very important for the task, and we took multiple precautions to ensure that all participants could see the objects adequately. First, kinetic perimetry was performed on each subject in order to precisely delineate the extent of their visual field. These data allowed us to make sure that participants were able to detect small stimuli across the entire screen. Secondly, the practice run allowed participants to voice any difficulties they encountered before starting the experiment. No single subject complained about the visibility of items.

Moreover, we would like to point out that it is nearly impossible to screen out participants who wear glasses when conducting an experiment on aging. Older adults who took part in our experiment were recruited from a cohort of healthy participants with no ophthalmological conditions. Notwithstanding, all older volunteers still required some form of correction (20/20 subjects), and more specifically progressive lenses. The latter are designed so that the top portion of the lens can be used for objects in the distance and that the bottom portion may be used to objects in very close proximity. Considering that we were interested in perceptual asymmetries along the vertical meridian, keeping the glasses on would have introduced a significant source of bias.

We did include analyses in the manuscript looking at the possible influence of visual capacity on task performance. No significant influence of visual acuity ($R^2 = 0.011$, $F(2, 17) = 0.099$, $p = 0.91$) or contrast sensitivity ($R^2 = 0.016$, $F(2, 17) = 0.14$, $p = 0.87$) was found on the age-related upper visual field deficit. To further convince the reviewer that the extent of the corrected visual deficit did not influence the results, we conducted a supplementary analysis. We categorized older adults as “*frequent*” or “*occasional*” wearers of progressive lenses. As progressive lenses distort the

visual field, we reasoned that wearing them more or less frequently may modify the way object information is encoded. We conducted a linear regression to investigate the impact of wearing frequency on parameter S_{up} in older adults. We found no significant difference in the probability of remembering the position of an item presented in the upper field (S_{up}) between frequent and occasional wearers of progressive lenses ($R^2 = -0.089$, $F(2, 17) = 0.13$, $p = 0.73$).

Finally, it is important to remind ourselves that parameters I_{up} and I_{down} , which reflect familiarity-based memory processes, were not significantly different between young and older adults. Both age groups were thus capable of recognizing items to a similar extent. In our opinion, this result is a clear indicator that poor vision does not explain the upper field spatial memory deficit in older adults.

2 -- I have concerns about the way the MPT has been specified. For one, the model is not described in sufficient detail. For example, parameters o and g are described with respect to old items, but not with respect to new items. This is consequential because the parameters appear in both old and new decision trees, but their meaning differs across them. This issue was discussed by Kellen and Singmann (2017) in a comment to the Cooper et al. (2017) article the authors cite. On the one hand, the description of the o and g parameters are inaccurate in the text – they do not describe the probabilities of correctly guessing “old” or “up” (for stimuli appearing in the upper position), as stated. If this were true, these parameters could not appear in the “New” decision tree because these branches all lead to errors. More fundamental is that the o parameter in the “New” tree refers to the probability of falsely recognizing a new item as old, conditioned on the failure of recollection and the subsequent failure of familiarity-based recognizing. That the parameters perform different duties depending on the stimulus makes them difficult to interpret because they are not characterizing the same psychological process.

We thank the reviewer for this comment and for discussing the comment by Kellen and Singmann (2017, *Cortex*). We realized that we wrongly defined parameters o and g in the original manuscript, which may have led to some confusion. Parameter o actually corresponds to the probability of guessing that the object was presented (i.e., participant answered “old”). Parameter g , on the other hand, estimates the probability of guessing that the item was presented in the upper visual field. In that respect, we wish to highlight that parameters o and g have the exact same meaning in both the old and new decision trees. The models in the Cooper et al. (2017a, *Cortex*) study that Kellen and Singmann rightly point out as being polysemous are the extended MPT models that include additional confidence judgments. However, in their comment, they fail to mention the simpler models developed by Cooper and colleagues (i.e., the *Item-Source* and *Source-Item* models) within which each parameter has a unique meaning. Our MPT model is equivalent to their *Source-Item* model and is in fact monosemous. The parameters in our MPT model perform the same duties irrespective of the stimulus (Old up, Old down, New). Cooper and colleagues have published a reply to the comment written by Kellen and Singmann that is very comprehensive (Cooper et al., 2017b, *Cortex*).

We amended the definitions for parameters g and o throughout the manuscript.

Methods section

- Lines 490 – 492: “The parameter o corresponds to the probability of guessing that the object was presented (i.e., answering “old”), and the parameter g estimates the probability of guessing that the item was presented in the upper part of the screen, given that item recognition has failed.”

Results section

- Lines 139 – 141: “We found that older adults’ probability of guessing that an item was presented during the encoding phase was significantly lower than young adults’.”
- Lines 204 – 205: “the guessing parameter g , defined as the probability of guessing that an item was presented in the upper visual field, was equivalent in young and older subjects.”
- Figure 2: “Parameter \$o\$ refers to the probability of guessing that an item is “old” while parameter \$g\$ refers to the probability of guessing that an item was presented in the upper visual field.”

Extended data section

- Figure E1. “The parameter o describes the probability of guessing that the item was presented (i.e., answering “old”). The parameter g on the other hand describes the probability of guessing that the item was presented in the upper visual field.”

3 -- It is unclear how the data were modelled using the MPT. Were the individual participant data modelled, and the resulting posterior estimates compared across groups, or were the data modelled hierarchically? The latter constitutes the best practice, but either way, some clarification is needed.

We agree with the reviewer that we should have better specified how the data were modeled. To fit our MPT model, we used the TreeBUGS R package (Heck et al., 2018, *Behav. Res. Methods*). This package was chosen on the grounds that it performs hierarchical modeling. This type of modeling assumes different parameters for each participant that follow a hierarchical distribution at the population-level. TreeBUGS has the additional advantage of being able to fit latent-trait hierarchical models. Not only do these models consider interindividual variability but they also recognize that separate parameters may be correlated within subjects. In the same vein, we realized that we should have been more explicit as to how parameter differences were evaluated between age groups.

We modified the Methods and Results sections in order to clarify the way the data were modeled.

Methods section:

- Lines 493 – 497: “We fit the behavioral data to the MPT model for each age group separately with the TreeBUGS package in R. This R package allows hierarchical models to be fit using Just Another Gibbs Sampler (JAGS) to estimate posterior distributions. Specifically, we used latent-trait hierarchical models because they take into account both interindividual variability and parameter correlation within subjects⁸².”
- Lines 507 – 510: “To test for differences in parameters between young and older participants, we subtracted the posterior distributions of each parameter of the older group from the young group (Fig. E2). We obtained posterior distributions of the age differences from which we extracted the mean and 95% credibility interval.”

Results section:

- Lines 93 – 94: “We first evaluated fits of the latent-trait hierarchical MPT model with posterior predictive p -values.”

4 -- The comparisons of the I_{Up} and I_{Down} parameters don’t tell the whole story. While performance is similar in the lower position, the evidence for no difference in the upper position is less convincing – credible intervals fail to overlap 0 by the narrowest of margins. Perhaps model selection might provide a better test – does a model with parameters

constrained to be equal across groups outperforms the one presented by the authors? My sense, however, is that the question of importance is whether young and older participants show similar patterns of performance across the two positions rather than absolute levels of performance within each position. That is, the difference between I_{Up} and I_{Down} within each participant group would be a more meaningful comparison to make across groups (viz. are the asymmetries comparable across age groups). A model selection way of formulating the question would set the parameter differences to be equal across groups. The critical comparison is whether this constrained model is outperformed by one where different parameter differences are required for older and younger groups. (Similar comments apply to the comparisons of S_{Up} and S_{Down}).

We are grateful for the reviewer's analysis suggestion. We agree that investigating how the differences between I_{up} and I_{down} as well as between S_{up} and S_{down} compare across age groups would elucidate some aspects of our data.

First, we wish to highlight that in studies using Bayesian hierarchical models it is most common to only look at differences in parameters between age groups by subtracting the posterior distributions (Bartsch et al., 2018, *Psychol. Aging*; Greene & Naveh-Benjamin, 2020, *Psychol. Sci.*; Smith & Batchelder, 2010, *J. Math. Psychol.*). The model selection approach suggested by the reviewer requires to fit the data in a single extended MPT model that pools young and older adults' data. The latter has been criticized for several reasons. Goodness of fit measures will tend to be disproportionately influenced by the group with the more aggregate numbers if sample sizes are unequal. In our experiment, there were twenty-six young adults and twenty older adults. Moreover, participant heterogeneity is not taken into consideration. Nevertheless, we have decided to conduct the suggested model selection analysis, and we included it in the Supplementary Information file.

In order to compare the parameter differences between age groups, we needed to sum trial counts across young and older subjects and fit the aggregate data in a single MPT model. This extended model comprised the three trees of the *Source-Item* model, doubled for each age group. Each parameter was thus added a label *Y* or *O* depending on whether it belonged to the young or older group. A schematic illustration of the new MPT model can be found in Figure 1. The reviewer suggests that we compare parameter asymmetries between age groups. To date, there are no R packages that allow equality constraints on parameter differences to be implemented (i.e., by setting the difference between I_{up} and I_{down} or S_{up} and S_{down} to be equal across groups). To circumvent this issue, we used order constraints to test interactions between various factors in our model (Schmidt et al., 2022, *Psychol. Methods*), and we tested whether the probabilities of object retrieval (*I*) and object position retrieval (*S*) varied as a function of item position (up vs. down) and age group (young vs. older).

Figure 1. Graphical representation of the extended MPT Source-Item model used to analyze aggregated data from young and older participants. A tree is used to model experimental data. Colored rounded rectangles represent the three different trial types: the object was presented in the upper part of the screen (“UP”), in the lower part of the screen (“DOWN”), or was not presented (“NEW”). Each tree is doubled as there is one stimulus type per age group in the extended MPT model. Grey rounded rectangles represent participants’ possible answers. I_{up} and I_{down} are the probabilities of remembering an item that was presented in the upper visual field and lower visual field respectively. S_{up} and S_{down} are the probabilities of remembering the position of an item that was presented in the upper visual field and lower visual field respectively. Parameter o refers to the probability of guessing that an item is “old” while parameter g refers to the probability of guessing that an item was presented in the upper visual field. The “Y” and “O” labels indicate whether the parameter is being estimated from the young or older adult group, respectively.

Order constraints can be implemented using reparameterization of the MPT model (Kuhlmann et al., 2019, *Front. Psychol.*). To this end, we assumed that the probabilities of item memory and of spatial memory are higher in young adults than in older adults ($I_O < I_Y$; $S_O < S_Y$). Building on these order constraints, four new auxiliary parameters α were created such that:

$$I_{down_O} = I_{down_Y} * \alpha I_{down}$$

$$I_{up_O} = I_{up_Y} * \alpha I_{up}$$

$$S_{down_O} = S_{down_Y} * \alpha S_{down}$$

$$S_{up_O} = S_{up_Y} * \alpha S_{up}$$

Equations were updated accordingly in the extended MPT model. Analyses were conducted in multiTree (Moshagen, 2010, *Behav. Res. Methods*), a software that allows for automatic reparameterization of MPT models. We first needed to establish a baseline model against which the order-constrained models could be compared. Considering that the extended model comprised

more parameters, we included the equality $g_Y = g_O$ for the baseline model to be identifiable. The baseline model was well fit ($G^2(1) = 1.59, p = 0.21$).

We then constructed two order-constrained models, one in which the auxiliary parameters related to item memory were set to be equal ($\alpha_{I_{down}} = \alpha_{I_{up}}$) and one in which the auxiliary parameters related to spatial memory were set to be equal ($\alpha_{S_{down}} = \alpha_{S_{up}}$). If adding these order constraints didn't modify model fits significantly, then we could conclude that the interaction between object position and age doesn't influence item or spatial memory. The ΔG^2 difference test revealed that the order-constrained model for item memory fit the data significantly worse than the baseline model ($\Delta G^2(1) = 10.94, p = 0.00094$). Similarly, the ΔG^2 difference test showed that the order-constrained model for spatial memory fit the data significantly worse than the baseline model ($\Delta G^2(1) = 6.07, p = 0.014$). From these results we can conclude that parameter asymmetries differ between young and older adults. According to the extended MPT model, the asymmetries between I_{down} and I_{up} and those between S_{down} and S_{up} differ in young and healthy older adults.

Results from the reparameterized MPT model are in line with our finding that S_{up} differs between young and older adults and that S_{down} doesn't. They also shed new light on differences between I_{down} and I_{up} . While our initial *Source-Item* model found that the two item memory parameters were equivalent between young and older adults, this extended model uncovers that item memory varies as a function of object position and age. It appears that the difference between I_{down} and I_{up} is more important in young adults than in older adults. Young adults show a greater lower visual field advantage for item recognition than do older adults. This finding resonates with the standard *Pr* analysis which uncovered an overall lower field bias for item memory. It seems reasonable to suggest that the influence of object position on item memory is driven by the younger group. Notwithstanding, these results are to be taken lightly as the unbalanced sample sizes between age groups could constitute an important source of bias.

We added the methods and results for this new analysis to the Supplementary Information file. We also modified the text to account for the fact that the credible interval for I_{down} fails to overlap 0 by a small margin, as noted by the reviewer.

Methods section

- Lines 510 – 513: “An alternative approach to evaluating group differences consists in summing trial frequencies across young and older participant and fitting the aggregate data in a single extended MPT model. The methods and results for this complementary analysis are available in the Supplementary Information File.”

Results section

- Lines 100 – 103: “As a complementary analysis, we also tested whether item memory and source memory performance varied as a function of object position and age group in an extended MPT model with aggregate data (Supplementary Information file).”
- Lines 138 – 139: “One may nonetheless note that the 95% BCI for I_{down} fails to overlap 0 by a narrow margin.”

Supplementary Information file

- Lines 1 – 37: “We conducted a complementary analysis in order to compare MPT parameter differences between young and older participants. In other words, we wished to evaluate how the difference between I_{down} and I_{up} as well as between S_{down} and S_{up} changed across age groups. For this purpose, we summed trial frequencies across young and older participants and fit the aggregate data into a single extended MPT model. The latter comprised the three trees of the *Source-Item* model, corresponding to the three stimulus types, doubled for each age group.

Accordingly, each parameter was added a label “Y” or “O” depending on whether it belonged to the young or older group, respectively. A schematic illustration of this new MPT model can be found in **Fig. S1**. Order constraints were used on the parameters of the model to investigate how item position and age group influenced item and spatial memory. To disentangle the main effects of these two factors and their interaction, we implemented the order constraints using reparameterization^{1,2,3}. We assumed that the probabilities associated with item memory and spatial memory were higher in young adults than in older adults. Building on these order constraints, we created four auxiliary parameters and modified the model equations accordingly. The α auxiliary parameters were created such that:

$$I_{\text{down_O}} = I_{\text{down_Y}} * \alpha I_{\text{down}}$$

$$I_{\text{up_O}} = I_{\text{up_Y}} * \alpha I_{\text{up}}$$

$$S_{\text{down_O}} = S_{\text{down_Y}} * \alpha S_{\text{down}}$$

$$S_{\text{up_O}} = S_{\text{up_Y}} * \alpha S_{\text{up}}$$

We first established a baseline model against which the order-constrained model would be compared. The baseline model had to include an equality constraint for the model to be identifiable. In accordance, parameter g was set to be equal across groups ($g_Y = g_O$). The two restricted models were then constructed; one in which the auxiliary parameters related to item memory were set to be equal ($\alpha I_{\text{down}} = \alpha I_{\text{up}}$), and the other in which the auxiliary parameters related to spatial memory were set to be equal ($\alpha S_{\text{down}} = \alpha S_{\text{up}}$). We then applied the ΔG^2 difference test for equality constraints in order to determine whether the interaction between object position and age influenced item or spatial memory. This extended analysis was conducted in multiTree⁴, a software that allows for automatic reparameterization of MPT models. It is important to note that this approach to studying MPT parameter differences has been criticized. Indeed, goodness of fit measures tend to be disproportionately influenced by the group with higher aggregate numbers if sample sizes are unequal⁵. Moreover, participant heterogeneity is here not considered. We thus stress that the results are to be taken with caution.

The baseline MPT model fit the data well ($G^2(1) = 1.59, p = 0.21$). The ΔG^2 difference test revealed that the order-constrained model for item memory fit the data significantly worse than the baseline model ($\Delta G^2(1) = 10.94, p = 0.00094$). Regarding spatial memory, the ΔG^2 difference test also revealed that the restricted model fit the data significantly worse than the baseline model ($\Delta G^2(1) = 6.07, p = 0.014$). These results imply that the differences between I_{down} and I_{up} and between S_{down} and S_{up} are not equivalent between young and healthy older populations. According to these order-constrained models, object position and age interact to influence both item memory and spatial memory.”

5 -- The results are mainly interpreted through the lens of a relative performance deficit for spatial information in the upper visual hemifield. The general thrust of the manuscript seems at odds with the visual trends in the data shown in Figure 2. Numerically, at least, it would appear that both groups exhibit a familiarity-based item recognition deficit in the upper vs. lower field and a spatial recollection benefit for the upper field. Given the issues with the analyses discussed above, I am not confident that the conclusions are appropriately supported by the data.

Following another reviewer’s comment that more traditional analyses, alongside the MPT modeling, may facilitate comprehension, we decided to include the standard *Pr* analyses in the main manuscript. They were initially in the Supplementary Information file. We divided Figure 2 from the original manuscript into 3 new figures. The new Figure 2 (displayed below) now illustrates

only the MPT binary trees that were used to model data. The updated Figures 3 and 4 (reported below) show results from the standard and MPT analyses related to item memory and spatial memory, respectively. We believe that these analyses, as well as the extended MPT model presented above, strengthen our conclusions.

• New Figure 2:

Figure 2. Graphical representation of the MPT model used to analyze the experimental data. A tree is used to model experimental data. Colored rounded rectangles represent the three different trial types: the object was presented in the upper part of the screen (“UP”), in the lower part of the screen (“DOWN”), or was not presented (“NEW”). Grey rounded rectangles represent participants’ possible answers. I_{up} and I_{down} are the probabilities of remembering an item that was presented in the upper visual field and lower visual field, respectively. S_{up} and S_{down} are the probabilities of remembering the position of an item that was presented in the upper visual field and lower visual field, respectively. Parameter o refers to the probability of guessing that an item is “old”, while parameter g refers to the probability of guessing that an item was presented in the upper visual field.

• New Figure 3:

Figure 3. Behavioral performance on the item memory task as evaluated by **(A)** standard Pr analyses and **(B)** MPT analyses. **(A)** Item memory is evaluated during the object recognition task in which subjects answer the question “Is the object old or new?”. Item Pr corresponds to the percentage of good answers minus the percentage of false alarms. Horizontal colored lines correspond to individual data points. Horizontal black lines correspond to the mean in each group. $*p < 0.05$ **(B)** Group-level posterior distributions for parameters related to item memory. The posterior distribution corresponds to updated knowledge about the parameters I_{up} and I_{down} after considering the current data. I_{up} is the probability of remembering an item that was presented in the upper part of the screen and I_{down} is the probability of remembering an item that was presented in the lower part of the screen.

- New Figure 4:

Figure 4. Behavioral performance on the spatial memory task as evaluated by **(A)** standard *Pr* analyses and **(B)** MPT analyses. **(A)** Spatial memory is evaluated during the spatial memory task in which subjects answer the question “Was the object presented in the upper or lower visual field?”. Source *Pr* corresponds to the percentage of correct answers weighted by the number of answered trials. Horizontal colored lines correspond to individual data points. Horizontal black lines correspond to the mean in each group. * $p < 0.05$. **(B)** Group-level posterior distributions for parameters related to spatial memory. The posterior distribution corresponds to updated knowledge about the parameters S_{up} and S_{down} after considering the current data. S_{up} is the probability of remembering the position of an item that was presented in the upper part of the screen, and S_{down} is the probability of remembering the position of an item that was presented in the lower part of the screen.

The reviewer remarks that both groups appear to be better at familiarity-based item recognition in the lower visual field. The standard *Pr* analyses do indeed reveal that item recognition benefits from a slight lower visual field advantage across all participants ($F(1, 44) = 11.58, p = 0.0014$). Interestingly, results from the extended MPT model showed that the interaction between object position and age had an effect on item memory (see **Comment 4**). In light of these findings, it is reasonable to speculate that the lower field benefit for item memory is mostly driven by young adults, the benefit being more tenuous in the older group.

Furthermore, we don't think that our data support an upper field benefit for spatial recollection across participants, as suggested by the reviewer. It is very clear from the standard analysis that there is no influence of object position on source *Pr* (i.e., the measure of accuracy for spatial memory) in young adults. However, the interaction between age group and object position is significant ($F(1, 43) = 5.37, p = 0.025$), which indicates that older adults have significantly worse

spatial performance for items presented in the upper portion of the screen. The same interaction was uncovered in the extended MPT model (see **Comment 4**).

We thank the reviewer for their comment because these additional analyses have not only consolidated our initial conclusions, but they have also shed new light on the data. It appears important to discuss the lower field advantage in item recognition. Therefore, we added some information in the Discussion section in order for the reader to gain a more comprehensive view of the data.

Discussion section

- Lines 244 – 246: “The findings revealed a general lower field advantage for item recognition and an age-related decline in spatial memory for objects presented in the upper visual field only.”
- Lines 267 – 276: “An effect of the objects’ vertical position on item memory was found across all participants. Object recognition and reaction times were improved for items presented in the lower portion of the screen. In the literature, object processing tasks including letter identification, target localization, word discrimination, and face sex-categorization, have been conferred an upper visual field advantage^{25,26,49,50}. One possibility is that visual experience with specific stimuli shapes the formation of vertical visual field biases. For example, researchers are evoking the possibility that faces are better encoded in the upper visual field because they are situated on the upper portion of the body^{31,51}. That everyday objects are more frequently encountered in the lower visual field may explain why the encoding of items in our task was facilitated in the lower portion of the screen for all participants.”
- Lines 319 – 322: “The slight lower field benefit for item memory in young and older adults is in line with fMRI studies reporting that the lateral occipital visual area (LO), an object-selective region, has stronger BOLD responses to objects in the lower than in the upper visual field.⁷³”

Minor concerns:

6 -- Lines 287-288 – Presentation time for items should be specified as 2500 ms per item. Current wording could be taken to imply that 30 objects were presented in 2500 ms.

We have modified the text accordingly.

Methods section

- Lines 391 – 393: “The task presented 30 unique objects sequentially, for 2500 ms each, in the upper or lower part of the screen along the vertical meridian.”

7 -- Line 293 – Can you specify the nature of the pseudorandomization of old and new items at test?

We thank the reviewer for their question as we realized that the order of old and new items at test was actually chosen randomly and not pseudorandomly. The order nonetheless remained fixed for all participants. We made sure that both old and new items were well balanced across blocks in terms of object category and object area.

These precisions were added to the Methods section of the revised manuscript.

Methods section

- Line 399: “Items were ordered randomly, but that order remained fixed across participants.”

8 -- Lines 322-323 – “We excluded two older subjects from the visual field analyses because their kinetic perimetry data were not exploitable”. I was unsure what was meant by “exploitable” here.

For these two older participants, there were issues with formatting their kinetic perimetry data (due to a wrong setting to extract raw data from the device). We hope the reviewer now better understands what we meant by “exploitable”.

We added a sentence in the Methods section to clarify what we meant by “not exploitable”.

Methods section

- Lines 428 – 430: “We excluded two older subjects from the visual field analyses because their kinetic perimetry data were extracted in a wrong format. The comparison between their data and those from other participants was thus impossible.”

References

Bartsch, L. M, Loaiza, V. M. & Oberauer, K. Does limited working memory capacity underlie age differences in associative long-term memory?. *Psychol. Aging* (2019)

Cooper, E., Greve, A. & Henson, R. N. Assumptions behind scoring source versus item memory: Effects of age, hippocampal lesions and mild memory problems. *Cortex* (2017a)

Cooper, E., Greve, A. & Henson, R. N. Assumptions behind scoring source versus item memory impact on conclusions about memory: A reply to Kellen and Singmann’s comment (2017). *Cortex* (2017b)

Greene, N. R. & Naveh-Benjamin, M. A specific principle of memory: Evidence from aging and associative memory. *Psychol. Sci.* (2020)

Kellen, D. & Singmann, H. Memory representations, tree structures, and parameter polysemy: Comment on Cooper, Greve and Henson (2017). *Cortex* (2017)

Kuhlmann, B. G., Erdfelder, E. & Moshagen, M. Testing interactions in multinomial processing tree models. *Front. Psychol.* (2019)

Moshagen, M. multiTree: a computer program for the analysis of multinomial processing tree models. *Behav. Res. Methods* (2010)

Heck, D. W., Arnold, N. R. & Arnold, D. TreeBUGS: An R package for hierarchical multinomial-processing tree modeling. *Behav. Res. Methods* (2018)

Schmidt, O., Erdfelder, E. & Heck, D. W. Tutorial on multinomial processing tree modeling: how to develop, test, and extend MPT models. *Psychol. Methods* (2022)

Smith, J. B. & Batchelder, W. H. Beta-MPT: Multinomial processing tree models for addressing individual differences. *J. Math. Psychol.* (2010)

13th Apr 23

Dear Dr Arleo,

Your manuscript titled "The Vertical Position of Visual Information Conditions Spatial Memory Performance in Healthy Aging" has now been seen by our reviewers, whose comments appear below. In light of their advice I am delighted to say that we are happy, in principle, to publish a suitably revised version in Communications Psychology under the open access CC BY license (Creative Commons Attribution v4.0 International License).

We therefore invite you to revise your paper one last time to address the remaining concerns of our reviewers and a list of editorial requests. At the same time we ask that you edit your manuscript to comply with our format requirements and to maximise the accessibility and therefore the impact of your work.

EDITORIAL REQUESTS:

SUBMISSION INFORMATION:

OPEN ACCESS:

Communications Psychology is a fully open access journal. Articles are made freely accessible on publication under a [CC BY license](http://creativecommons.org/licenses/by/4.0) (Creative Commons Attribution 4.0 International License). This license allows maximum dissemination and re-use of open access materials and is preferred by many research funding bodies.

For further information about article processing charges, open access funding, and advice and support from Nature Research, please visit <https://www.nature.com/commspsychol/article-processing-charges>

At acceptance, you will be provided with instructions for completing this CC BY license on behalf of all authors. This grants us the necessary permissions to publish your paper. Additionally, you will be asked to declare that all required third party permissions have been obtained, and to provide billing

information in order to pay the article-processing charge (APC).

* **DATA AVAILABILITY:**

[link redacted]

Best regards,

Jennifer Bellingtier

Jennifer Bellingtier, PhD
Senior Editor
Communications Psychology

REVIEWERS' EXPERTISE:

Reviewer #1 visual attention, memory

Reviewer #2 visual attention, memory

Reviewer #3 memory, computational analysis

REVIEWERS' COMMENTS:

Reviewer #1 (Remarks to the Author):

I carefully read the authors' response to reviewers about the manuscript titled: "The Vertical Position of Visual Information Conditions Spatial Memory Performance in Healthy Aging".

I would like to underline how transparent and thorough the authors' responses were. They addressed each of my concerns (except one minor suggestion for which I can understand their choice, albeit less informative to me). In particular, following my major concern, they assessed the source expectation of their object stimuli and conducted two supplementary analyses to rule out the presence of evidence pointing to a bias relative to the overall pre-experimental source expectation. This is valuable, and greatly adds to the demonstration of the central claim. I further appreciated how they included their response in the manuscript, as well as the discussion they added about these analyses.

Overall, I believe this manuscript will make a useful contribution to the literature.

Minor comment:

- How the authors assigned the pre-experimental source expectation to each item included in the experiment (i.e., upper or lower expectation) should be briefly detailed. Did they use a method like in Kaiser et al. (2018, cited in their response)? Or was it assessed by one rater only? In the latter case, I would suggest that two raters (or even three, to use the most frequent values), along with an index validating the reproducibility, would yield a more robust estimation.

Reviewer #2 (Remarks to the Author):

The authors have appropriately addressed most of the concerns I raised in my initial review (attention, fixation, number of test items, estimates of guessing, etc.), either through additions or corrections to the paper, or through further explanations in their responses.

I can now endorse the paper for publication.

Reviewer #3 (Remarks to the Author):

My main concerns with the initial submission were based on problems I perceived with the way the MPT analysis was conducted. Indeed, there were some issues with how the guessing parameters were defined in the original manuscript.

I very much appreciate the care the authors took in responding to my earlier comments. The inclusion of an alternative way of analyzing the data, by aggregating data and comparing fits of nested MPTs is no small undertaking, and I think, overall, it complements their primary analysis

nicely.

Given the modifications the authors have made to the models, and their comprehensive responses to my other comments, I have no other major concerns. I therefore recommend the manuscript for publication, given that it addresses an interesting question with a principled analytic method.